# Mycobacterial metallophosphatase MmpE acts as a nucleomodulin to regulate host gene expression and promote intracellular survival

Liu Chen[1,2], Baojie Duan[1], Qiang Jiang[1], Yifan Wang[1], Yingyu Chen[1,3], Lei Zhang[1,3]*, Aizhen Guo[1,2,3]*

[1]National Key Laboratory of Agricultural Microbiology, College of Veterinary Medicine, Huazhong Agricultural University, Wuhan, China; [2]Hubei Hongshan Laboratory, Huazhong Agricultural University, Wuhan, China; [3]National Professional Laboratory for Animal Tuberculosis, Ministry of Agriculture and Rural Affairs, Huazhong Agricultural University, Wuhan, China

*For correspondence:
zhanglei2023@mail.hzau.edu.cn (LZ);
aizhen@mail.hzau.edu.cn (AG)

## eLife Assessment

The work **convincingly** demonstrates the role of the mycobacterial secreted effector protein MmpE, which translocates to the host nucleus and exhibits phosphatase activity. The study is particularly **valuable** in showing that both the nuclear localization signal sequences and residues critical for phosphatase function are essential for host gene regulation, lysosomal biogenesis, and intracellular survival. Future studies will be needed to explore additional host pathways modulated by MmpE, particularly in the context of infection with a fully virulent *Mycobacterium tuberculosis* strain.

**Abstract** *Mycobacterium tuberculosis*, the causative agent of tuberculosis, remains a major global health challenge. Nucleomodulins, bacterial effectors that target the host cell nuclei, are increasingly recognized as key virulence factors, but their roles in mycobacterial pathogenesis remain incompletely elucidated. Here, we characterize a hypothetical protein Rv2577 (designated MmpE) not only as a $Fe^{3+}/Zn^{2+}$-dependent metallophosphatase but also as a critical nucleomodulin involved in immune evasion and intracellular persistence. MmpE utilizes two nuclear localization signals, RRR[20-22] and RRK[460-462], to enter the host cell nucleus, where it binds to the promoter region of the vitamin D receptor (VDR) gene, thereby inhibiting host inflammatory gene expression. Additionally, MmpE regulates the PI3K-Akt-mTOR signaling pathway, thereby arresting lysosome maturation. These actions collectively facilitate immune suppression and promote mycobacterial survival in macrophages and in mice. Our findings identify MmpE as a conserved nucleomodulin in mycobacteria and reveal a novel mechanism of MmpE-mediated intracellular survival.

## Introduction

*Mycobacterium tuberculosis* (Mtb), the etiological agent of tuberculosis, has posed a persistent global health burden for centuries. Despite significant advances in elucidating the molecular mechanisms of its pathogenesis, current research has largely focused on effector proteins that modulate host cell membrane and cytoplasmic processes (*Nisa et al., 2022*; *Nasiri and Venketaraman, 2025*). In

contrast, investigations into effector proteins that target the host cell nucleus, the central hub for gene regulation and cellular control, remain comparatively limited.

Nucleomodulins are a class of bacterial effector proteins that can translocate into the host nucleus, where they modulate nuclear processes to promote pathogen survival (*Chai et al., 2020*; *Bierne and Cossart, 2012*). These proteins affect host gene expression by directly interacting with chromatin or by mimicking transcription factors, chromatin modifiers, or other nuclear regulators (*Li et al., 2013*; *Rolando et al., 2013* ; *Mitra et al., 2018*; *Hanford et al., 2021*). For instance, LntA from *Listeria monocytogenes* directly binds the chromatin repressor BAHD1, disrupting its function and derepressing interferon-stimulated genes in a type III interferon-dependent manner, ultimately promoting bacterial persistence (*Lebreton et al., 2011*; *Lebreton et al., 2014*). Similarly, Ank1 and Ank6 from *Orientia tsutsugamushi* localize to the host nucleus and facilitate the export of the NF-κB p65 subunit via exportin 1, thereby suppressing the transcription of pro-inflammatory genes (*Evans et al., 2018*; *Steiert and Weber, 2025*). Although bacterial nucleomodulins utilize diverse strategies to manipulate host nuclear functions, the identity and functional characterization of *M. tuberculosis* nucleomodulins remain poorly defined.

Nucleomodulins employ various strategies to enter access to the host cell nucleus, including passive diffusion, hijacking host proteins that contain nuclear localization signals (NLSs), or using their own NLSs to interact with the nuclear pore complex (*Hanford et al., 2021*). Among these mechanisms, NLSs play a particularly important role. Classical NLSs are further divided into monopartite (MP) and bipartite (BP) forms. Monopartite NLSs consist of a single cluster of 4–8 basic residues, primarily lysine (K) or arginine (R), with a typical consensus motif of K-K/R-X-K/R (*Fontes et al., 2000*; *Kosugi et al., 2009*; *Cautain et al., 2015*). Nuclear localization enables these effectors to manipulate host gene expression, disrupt immune signaling, and reprogram cellular processes to promote pathogen survival. For example, EspF from enteropathogenic *Escherichia coli* contains an N-terminal NLS that directs it to the nucleolus, where it interferes with ribosome biogenesis (*Dean et al., 2010*; *Singh et al., 2018*). NUE, a SET domain-containing nucleomodulin secreted by *Chlamydia trachomatis*, also possesses an NLS that mediates its nuclear localization, where it functions as a histone lysine methyltransferase (HKMTase) targeting host histones H2B, H3, and H4 (*Pennini et al., 2010*; *Jermy, 2010*).

Beyond their nuclear localization capabilities, some nucleomodulins directly affect host immunity through intrinsic enzymatic activities (*Agarwal et al., 2012*). For example, the nucleomodulin PtpA promotes Mtb survival within macrophages by dephosphorylating host cytoplasmic proteins (*Wang et al., 2017*). NleC from *E. coli*, a Zn-dependent metalloprotease, suppresses host immune responses by cleaving the p65 subunit of NF-κB in the cytoplasm, thereby inhibiting its nuclear translocation, and can enter the nucleus to reduce nuclear p65 levels, blocking NF-κB-dependent transcription of pro-inflammatory genes (*Baruch et al., 2011*; *Hodgson et al., 2015*). Among these enzymatically active nucleomodulins, purple acid phosphatases (PAPs) represent a distinct and functionally significant, yet underexplored, subclass within the metallophosphatase superfamily. PAPs are characterized by a conserved β-α-β-α-β fold and five signature motifs (DxG, GDXXY, GNH[D/E], VXXH, GHXH), which coordinate seven metal-ligating residues at a binuclear metal center (*Rodriguez et al., 2014*; *Bhadouria et al., 2017*; *Bhadouria and Giri, 2022*). These enzymes have been extensively studied in plants and mammals, where they are involved in key biological processes, such as phosphorus metabolism and the generation of reactive oxygen species (*Comba et al., 2013*; *Feder et al., 2020*; *Liu et al., 2025*). However, only a few bacterial PAPs have been identified, such as BcPAP from *Burkholderia cenocepacia*, and their functions remain largely unknown (*Yeung et al., 2009*). Notably, in Mtb, MmpE is the only identified PAP protein (*Schenk et al., 2000*), but its specific contribution to infection and pathogenesis remains elusive.

In this study, we characterized MmpE as a conserved nucleomodulin in mycobacteria and demonstrated its nuclear translocation mediated by two functional motifs. Nuclear-localized MmpE regulates host immune responses by repressing the vitamin D receptor (VDR), a key regulator of antimicrobial gene expression, and disrupting lysosome maturation through suppression of the PI3K-Akt-mTOR signaling pathway. Additionally, MmpE exhibits $Fe^{3+}$-dependent metallophosphatase activity, contributing to intracellular survival. These findings establish MmpE as a bifunctional effector that integrates immune modulation and enzymatic catalysis, underscoring its role in mycobacterial pathogenesis and potential as a therapeutic target.

## Results

### NLSs are required for the nuclear translocation of MmpE

In a prior study, we screened conserved nucleomodulins in Mtb and found that hypothetical protein Rv2577 (MmpE) exhibits strong nuclear localization in HEK293T cells (*Chen et al., 2025*). Furthermore, we observed a time-dependent increase in nuclear MmpE-EGFP levels in HEK293T cells (*Figures 1A*). Nuclear-cytoplasmic fractionation assays further confirmed these results, showing a progressive enrichment of MmpE in the nucleus during the course of transfection (*Figure 1B*), suggesting that MmpE might possess a nuclear translocation ability.

MmpE contains two putative nucleus localization signals, NLS1 (RRR$^{20-22}$) and NLS2 (RRK$^{460-462}$), which are located in the loop regions of the MmpE structure predicted by AlphaFold (*Figure 1C*). Deleting either the individual NLS or both simultaneously does not induce a conformational change in the core structure (*Figure 1—figure supplement 1B*). To assess the effects of NLSs on nuclear localization, we constructed NLS-deleted mutants, including MmpE$^{\Delta NLS1}$, MmpE$^{\Delta NLS2}$, and MmpE$^{\Delta NLS1-2}$ and transfected them into HEK293T cells. Fluorescence microscopy at 36 hr post-transfection (hpt) revealed that deletion of either NLS significantly reduced nuclear accumulation of MmpE-EGFP, while simultaneous deletion of both NLS motifs (MmpE$^{\Delta NLS1-2}$) completely abolished nuclear localization, resulting in exclusive cytoplasmic distribution (*Figure 1D and E*). Consistent with the microscopy results, nuclear-cytoplasmic fractionation showed that WT MmpE was present in detected both cytoplasmic and nuclear fractions, whereas MmpE$^{\Delta NLS1}$ and MmpE$^{\Delta NLS2}$ exhibited substantially reduced nuclear levels. Notably, the double mutant MmpE$^{\Delta NLS1-2}$ was completely undetectable in the nucleus (*Figure 1F and G*), suggesting that both NLS motifs are necessary for nuclear import.

To further detect whether MmpE translocates into the host nucleus during mycobacterial infection, we conducted recombinant *M. bovis* BCG strains expressing C-terminally Flag-tagged WT MmpE or the NLS deletion mutants, including MmpE$^{\Delta NLS1}$, MmpE$^{\Delta NLS2}$, and MmpE$^{\Delta NLS1-2}$. Immunoblot analysis of culture filtrates showed that both MmpE-Flag and the secreted antigen Ag85B were detectable in the supernatant of wild-type strains, whereas secretion of MmpE$^{\Delta NLS1}$ or MmpE$^{\Delta NLS1-2}$ were barely detectable, suggesting that NLS1 are required for the efficient secretion of MmpE (*Figure 1—figure supplement 1C*). Following infection of THP-1 macrophages, nuclear-cytoplasmic fractionation revealed that wild-type MmpE localized to both cytoplasmic and nuclear compartments, while MmpE$^{\Delta NLS1}$, MmpE$^{\Delta NLS2}$, or MmpE$^{\Delta NLS1-2}$ were almost undetectable in nuclear fractions (*Figure 1—figure supplement 1D*). These findings indicate that nuclear localization of MmpE depends on two distinct NLS motifs, with NLS1 being essential for effective secretion.

### Deletion of NLSs does not alter the phosphatase activity of MmpE

MmpE is a putative Fe$^{3+}$/Zn$^{2+}$-metallophosphatase with a purple acid phosphatase (PAP) motif (*Figure 2A*; *Figure 2—figure supplement 1A*). Multiple sequence alignment and phylogenetic reconstruction revealed that MmpE is highly conserved across mycobacterial species and harbors canonical characteristics of metallophosphatases with GD, GNHX, and GHXH motifs (*Figure 2B*, *Figure 2—figure supplement 1B*). These motifs are typically associated with enzymes that catalyze phosphate hydrolysis via a binuclear metal ion-dependent mechanism, wherein residues, such as Asp, His, and Asn coordinate essential metal cofactors, commonly Fe$^{3+}$, Mn$^{2+}$, or Zn$^{2+}$, at the active site. To test this, recombinant MBP-tagged MmpE without the Tat signal peptide (MmpE$^{\Delta Tat}$) was purified and subjected to in vitro phosphatase assays using *p*-nitrophenyl phosphate (*p*-NPP) as the substrate. Different concentrations of Fe$^{3+}$ or Zn$^{2+}$ were added to the reactions to detect their effects on the enzyme activity of MmpE. At the same ion concentrations, Fe$^{3+}$ is more effective than Zn$^{2+}$ in stimulating MmpE hydrolysis of *p*-NPP. The maximal enzymatic activity was observed at 50 μM Fe$^{3+}$ (*Figure 2C*). To identify the key residues of MmpE involved in metal coordination, a molecular docking experiment was performed using PyMOL on the structure of MmpE predicted by AlphaFold. His348 and Asn359 are supposed to be involved in metal ion coordination within the active site (*Figure 2D*). Then, the double mutant MmpE$^{\Delta Tat}$-H348AN359A was conducted to verify the essential residues for enzyme activity. As shown in *Figure 2E*, the addition of 50 μM Fe$^{3+}$ significantly increased the in vitro phosphatase activity of MmpE$^{\Delta Tat}$ compared to reactions without metal ions. However, the enzymatic activity of the MmpE$^{\Delta Tat}$-H348AN359A was significantly reduced under the same conditions, indicating that the residues H348 and N359 are essential for metal binding and catalytic activity. Additionally, we found that deletion of the NLSs does not affect the

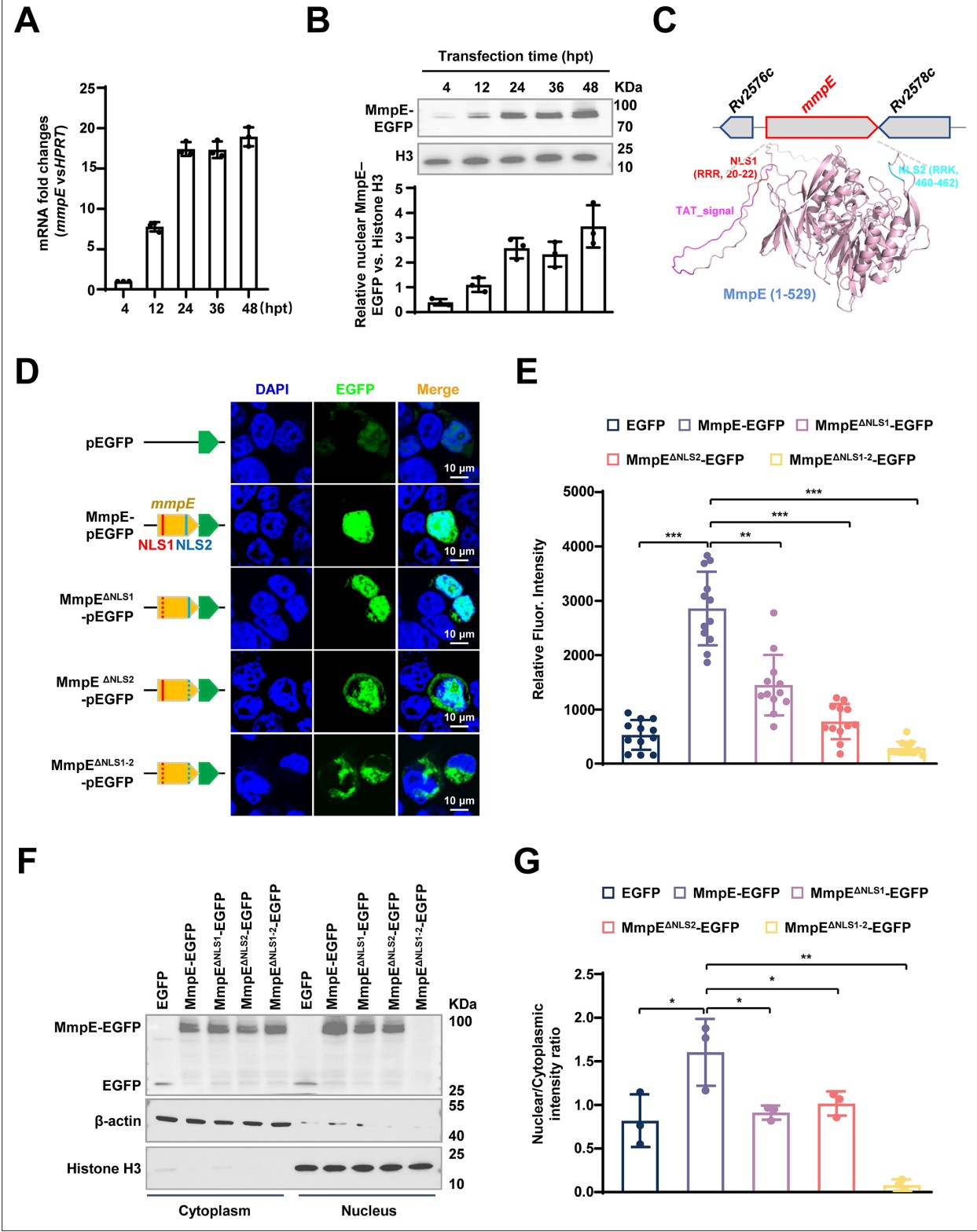

**Figure 1.** Nuclear localization signals (NLSs) are required for the nuclear translocation of MmpE. (**A**) qRT-PCR analysis of *mmpE* mRNA expression in HEK293T cells over 48 hpt. (**B**) Western blot analysis of nuclear fractions showing time-dependent accumulation of MmpE-EGFP (top), and corresponding quantification of nuclear MmpE-EGFP levels (bottom). Histone H3 and β-actin were used as nuclear and cytoplasmic markers, respectively. (**C**) Domain architecture of MmpE, including a Tat signal peptide (1–54 amino acids, with a twin-arginine translocation motif) and two nuclear localization signals (NLS1: 20–22 aa; NLS2: 460–462 aa). The structure of the MmpE protein was predicted using AlphaFold, with NLS1 and

*Figure 1 continued on next page*

*Figure 1 continued*

NLS2 highlighted in red and green, respectively. (**D**) Subcellular localization of EGFP-tagged wild-type and NLS-deleted MmpE. (Left) Schematic representation of EGFP-tagged constructs. (Right) Confocal microscopy images of HEK293T cells transfected with the indicated constructs for 36 hpt. EGFP fluorescence (green) and nuclear staining with DAPI (blue) were visualized using an FLUOVIEW software (v5.0). Scale bar, 10 μm. Images were acquired with a 100x oil immersion objective (NA = 1.4). (**E**) Quantification of nuclear EGFP intensity of wild-type and mutant MmpE in **D**. Data are shown as mean ± SD, n=12 cells. (**F**) Western blot analysis of nuclear and cytoplasmic fractions from HEK293T cells transfected with wild-type and mutant MmpE-EGFP confirmed their subcellular localization. MmpE-EGFP was detected using an anti-GFP antibody, and histone H3 and β-actin served as nuclear and cytoplasmic markers, respectively. (**G**) Quantitative analysis of MmpE-EGFP levels in the cytoplasmic and nuclear compartments of **F**. Data represent mean ± SD of three independent biological replicates. Statistical significance determined using two-tailed unpaired Student's *t*-tests, *$p<0.05$, **$p<0.01$, ***$p<0.001$.

The online version of this article includes the following source data and figure supplement(s) for figure 1:

**Source data 1.** Original western blots for panel B and F, indicating the relevant bands.

**Source data 2.** Original files for western blot analysis displayed in panel B and F.

**Figure supplement 1.** Identification of MmpE as a nucleomodulin in Mycobacterium.

**Figure supplement 1—source data 1.** Original western blots for panel C and D, indicating the relevant bands.

**Figure supplement 1—source data 2.** Original files for western blot analysis displayed in panel C and D.

phosphatase activity of MmpE (*Figure 2F*), consistent with the observation on comparison of the structures (*Figure 1—figure supplement 1B*). Collectively, these results indicate that MmpE is a highly conserved $Fe^{3+}$-dependent metallophosphatase in mycobacteria, and its enzymatic activity is independent of the NLSs.

## The nuclear translocation and phosphatase activity of MmpE are essential for *M. bovis* BCG survival in macrophage cells

To examine whether MmpE is important for bacterial intracellular survival, we constructed an *mmpE*-deletion mutant (ΔMmpE) in BCG strain by homologous recombination (*Figure 3—figure supplement 1A–C*). The ΔMmpE mutant and the ΔMmpE strains complemented with WT *mmpE* (Comp-MmpE) or NLS-deleted *mmpE* (Comp-MmpE$^{ΔNLS1}$, Comp-MmpE$^{ΔNLS2}$, and Comp-MmpE$^{ΔNLS1-2}$) exhibited similar growth rate in the standard 7H9 media (*Figure 3—figure supplement 1D*). During infection of THP-1 macrophages, the ΔMmpE strain exhibited a significantly reduced intracellular bacterial burden and accelerated clearance compared to the wild-type (WT) strain. This phenotype was restored by complementation with wild-type MmpE. Importantly, the load of the Comp-MmpE$^{ΔNLS2}$ displayed significantly lower compared to Comp-MmpE in THP-1 cells (*Figure 3A*). A comparable phenotype was also observed in RAW264.7 macrophages (*Figure 3B*). These findings indicate that NLS2-mediated nuclear localization is critical for the intracellular function of MmpE. Furthermore, the Comp-MmpE$^{ΔNLS1}$ and Comp-MmpE$^{ΔNLS1-2}$ strains showed partial restoration of bacterial load compared to the ΔMmpE strain but remained lower than those of the Comp-MmpE strain (*Figure 3C and D*). These results demonstrate that both NLSs are critical for MmpE-mediated bacterial survival within macrophages. Given that MmpE functions as a $Fe^{3+}$-dependent metallophosphatase in mycobacteria, we examined whether its enzymatic activity contributes to intracellular persistence and constructed a phosphatase-deficient double mutant strain (Comp-MmpE-H348AN359A) (*Figure 3—figure supplement 1E*) to infect THP-1 macrophage. The result showed that a significantly reduced intracellular bacterial burden compared to Comp-MmpE, indicating that the phosphatase activity also plays an important role in MmpE-mediated bacterial survival (*Figure 3E*). To further investigate the underlying mechanism, we analyzed cytokine expression in infected macrophages. Cells infected with the ΔMmpE and Comp-MmpE$^{ΔNLS1-2}$ strains exhibited significantly increased levels of pro-inflammatory cytokine, including *IL1A*, *IL1B*, and *IL6*, compared to the WT and Comp-MmpE strains. Furthermore, in comparison to the Comp-MmpE strains, cells infected with the Comp-MmpE$^{ΔNLS1}$, Comp-MmpE$^{ΔNLS2}$, and Comp-MmpEH348AN359A strains also demonstrated substantial upregulation of these cytokines (*Figure 3F–H*). Collectively, these results suggest that both nuclear localization and phosphatase activity are essential for the immunomodulatory function of MmpE, which facilitates bacterial survival by suppressing host inflammatory responses.

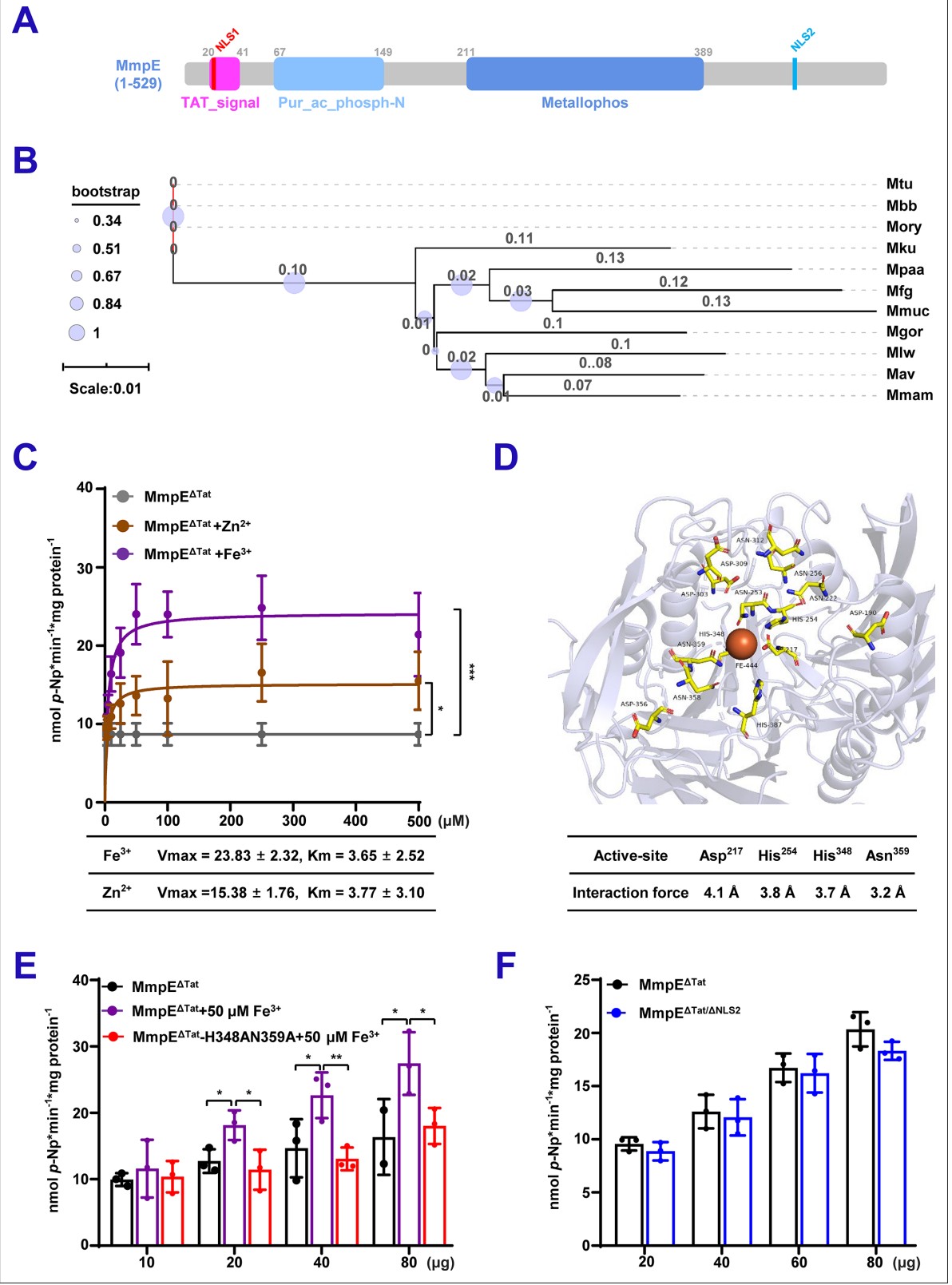

**Figure 2.** Deletion of nuclear localization signals (NLSs) does not alter the phosphatase activity of MmpE. (**A**) Domain architecture of MmpE. Schematic representation of MmpE with the annotated functional domains, including a Tat signal peptide (1–41 aa, twin-arginine translocation motif), a purple acid phosphatase domain (68–149 aa), a calcineurin-like phosphoesterase domain (211–389 aa) and two nuclear localization signals (NLS1: 20–22 aa; NLS2: 460–462 aa). (**B**) Phylogenetic and structural conservation of MmpE. Neighbor-joining phylogenetic tree of MmpE homologs across Mycobacterium

*Figure 2 continued on next page*

*Figure 2 continued*

species (1000 bootstrap replicates; values ≥ 50% shown). Species abbreviations: *M. tuberculosis* (Mtu), *M. bovis* BCG (Mbb), *M. orygis* (Mory), *M. kubicae* (Mku), *M. paraterrae* (Mpaa), *M. farcinogenes* (Mfg), *M. mucogenicum* (Mmuc), *M. vicinigordonae* (Mgor), *M. lentiflavum* (Mlw), *M. avium* (Mav), *M. manitobense* (Mmam). (**C**) Phosphatase activity of MmpE$^{\Delta Tat}$ (lacking the N-terminal Tat signal peptide) was measured using *p*-NPP as the substrate in the presence of increasing concentrations (0–500 μM) of $Fe^{3+}$ and $Zn^{2+}$. (**D**) Prediction of metal ion-binding residues in MmpE. Structural modeling and visualization were performed using PyMOL. Conserved residues in the putative metal-binding pocket are shown as sticks, colored by atom type. The predicted metal coordination site is highlighted, with key binding residues labeled. Surface representation depicts the spatial accessibility of the pocket. (**E**) Phosphatase activity of increasing concentrations of MmpE$^{\Delta Tat}$ and the mutant MmpE$^{\Delta Tat}$-H348AN359A was measured in the absence or presence of 50 μM $Fe^{3+}$. (**F**) Phosphatase activity of increasing concentrations of MmpE$^{\Delta Tat}$ and MmpE$^{\Delta Tat/\Delta NLS1-2}$ was measured under standard conditions. Data represent mean ± SD of three independent biological replicates. Statistical significance determined using two-way ANOVA, *$p<0.05$, **$p<0.01$, ***$p<0.001$.

The online version of this article includes the following figure supplement(s) for figure 2:

**Figure supplement 1.** Identification of MmpE as a conserved $Fe^{3+}/Zn^{2+}$-metallophosphatase in Mycobacteria.

## MmpE regulates host transcription network involved in inflammation response and lysosomal maturation

To investigate the effect of MmpE on host gene expression, we performed RNA-seq experiments on THP-1 macrophages infected with WT or ΔMmpE at 24 hpi. A total of 175 differentially expressed genes (DEGs) were identified, with 142 upregulated and 33 downregulated in ΔMmpE-infected cells compared to WT-infected cells (*Figure 4A*, *Supplementary file 1*). Gene ontology (GO) analysis revealed significant enrichment in immune-related biological processes, including apoptosis and cell death, molecular functions associated with immune activation and cellular stress responses, such as pyrophosphatase activity and nucleoside-triphosphatase activity (*Figure 4B*). KEGG pathway enrichment analysis showed that these DEGs were primarily involved in immune and inflammatory signaling pathways, including cytokine-cytokine receptor interaction, TNF, Toll-like receptor, chemokine, NF-κB, JAK-STAT, and PI3K-Akt signaling pathways. Additionally, a subset of DEGs was involved in lysosome-associated host pathways, consistent with previous findings that MmpE inhibits lysosomal maturation (*Forrellad et al., 2020*; *Figure 4C*). To further explore the functional relevance of the DEGs, we performed protein-protein interaction (PPI) network analysis focusing on protein-coding genes, which accounted for 38% of the total DEGs (*Figure 4—figure supplement 1A*). The results suggested that some core proteins, including IL23A, IL12B, CSF2, CD69, IDO1, and CEACAM1, are mainly related to the production of inflammatory cytokine and immune regulation (*Figure 4D*). Consistently, these genes showed increased expression in ΔMmpE-infected macrophages compared to those infected with wild-type BCG (*Figure 4E*). These results suggest that MmpE suppresses immune and lysosome-associated gene expression during infection.

Furthermore, we performed qRT-PCR to confirm the expression of key lysosomal maturation markers, including *TFEB*, *LAMP1*, *LAMP2*, and several V-ATPase subunit genes (*ATP6V0A1*, *ATP6V0C*, *ATP6V1A*, *ATP6V1B2*, and *ATP6V1E1*), as well as inflammatory cytokines. The expression levels of these genes were significantly higher in ΔMmpE-infected macrophages compared to those infected with WT (*Figure 4F and G*). Moreover, infection with the Comp-MmpE$^{\Delta NLS1-2}$ strain resulted in significantly increased gene expression compared to the Comp-MmpE strain (*Figure 4—figure supplement 1B–D*). Taken together, these results suggest that MmpE suppresses host immune and lysosomal responses to facilitate immune evasion during infection.

## MmpE regulates the PI3K-Akt-mTOR signaling pathway during macrophage infection

Given that MmpE modulates host transcriptional responses and functions as a nucleomodulin, we investigated whether it associates with host chromatin to regulate gene expression. Cleavage under targets and tagmentation (CUT&Tag) was performed in HEK293T cells transfected with MmpE-EGFP. A total of 2903 putative MmpE-binding sites were identified (*Supplementary file 2*), of which 18.49% (n=537) were located in intergenic regions and 39.44% (n=1145) in intragenic regions (*Figure 5A*). Chromosomal mapping revealed preferential enrichment on chromosomes 1 and 2 (*Figure 5B*). Notably, 99.7% (n=2894) of these binding sites were associated with protein-coding genes, and 1013 of them were located within 3 kb upstream of transcription start sites (TSSs), including 63.18% within 1 kb, 22.9% between 1–2 kb, and 13.92% between 2–3 kb (*Figure 5C and D*), suggesting that MmpE

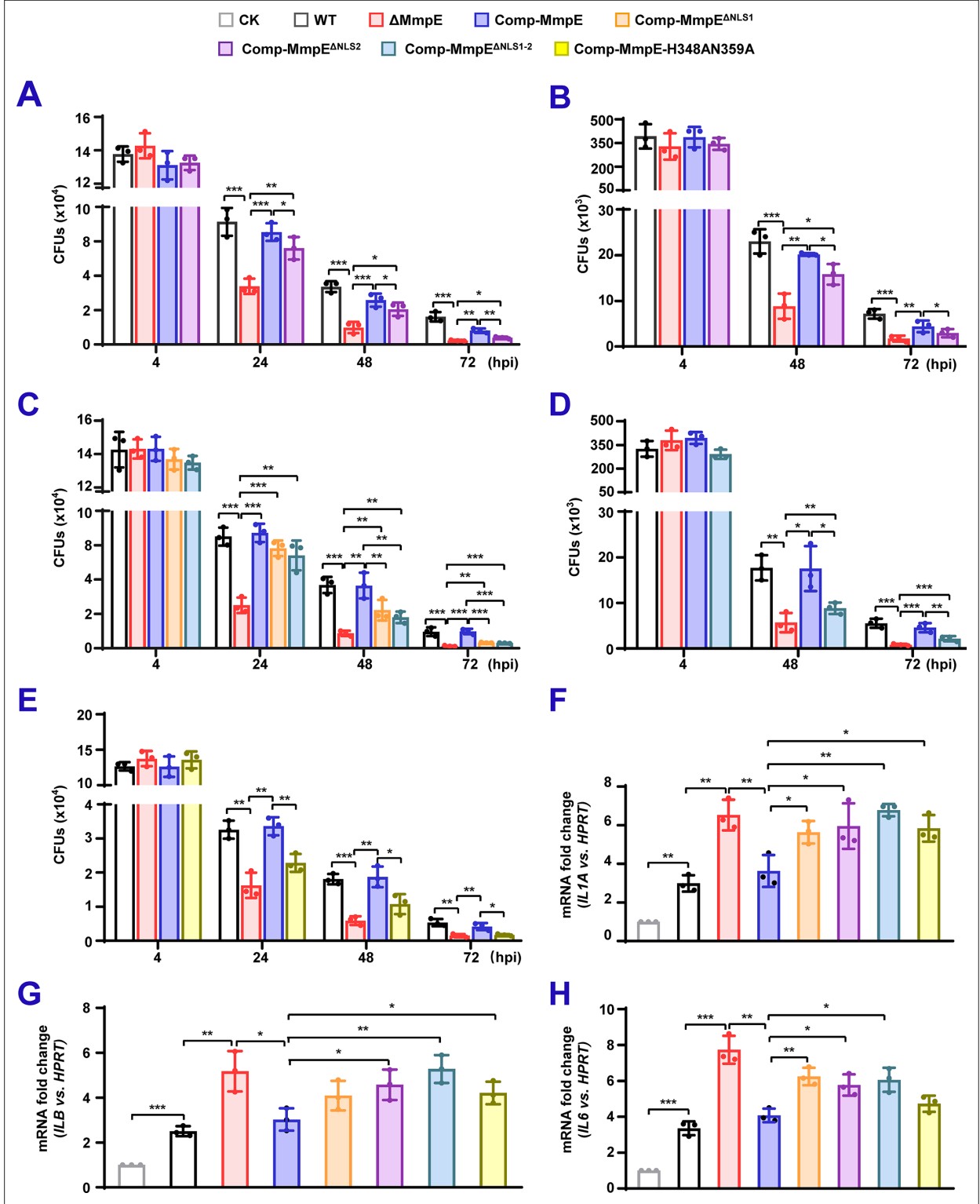

**Figure 3.** The nuclear translocation and phosphatase activity of MmpE are essential for *M. bovis* BCG survival in macrophage cells. (**A–B**) Intracellular survival of BCG strains in human THP-1 macrophages (**A**) and RAW264.7 macrophages (**B**). Strains include BCG wild-type (WT), ΔMmpE, Comp-MmpE, and Comp-MmpE^ΔNLS2. (**C–D**) Intracellular survival of strains in THP-1 (**C**) and RAW264.7 macrophages (**D**). Strains include BCG WT, ΔMmpE, Comp-MmpE, Comp-MmpE^ΔNLS1 and Comp-MmpE^ΔNLS1-2. (**E**) Intracellular survival of Comp-MmpE-H348AN359A mutant in THP-1 cells. (**F–H**) Inflammatory cytokine expression in infected THP-1 cells. mRNA levels of *IL-1A* (**F**), *IL-1B* (**G**), and *IL-6* (**H**) were quantified by qRT-PCR 24 hpi with the indicated BCG

*Figure 3 continued on next page*

Figure 3 continued

strains. Data represent mean ± SD of three independent biological replicates. Statistical significance determined using two-way ANOVA or two-tailed unpaired Student's *t*-tests, *$p$<0.05, **$p$<0.01, ***$p$<0.001.

The online version of this article includes the following source data and figure supplement(s) for figure 3:

**Figure supplement 1.** The nuclear translocation and phosphatase activity of MmpE are essential for *M. bovis* BCG survival in macrophage cells.

**Figure supplement 1—source data 1.** Original agarose gel images for panel A, indicating the relevant bands.

**Figure supplement 1—source data 2.** Original files for agarose gel electrophoresis displayed in panel A.

primarily targets promoter regions to modulate transcription. PPI analysis of the 1013 putative target genes revealed significant enrichment in immune-related kinase genes, including *PRKCB*, *PLCG2*, and *PIK3CB*, which are key regulators of inflammatory signaling and block lysosomal maturation via the PI3K-AKT-mTOR pathway (*Tian et al., 2020*; *Zheng et al., 2025*; *Figure 5—figure supplement 1A and B*). These findings are consistent with transcriptomic analysis (*Figure 4C*) and support the hypothesis that MmpE modulates host immune responses, potentially through the PI3K-AKT signaling axis. To validate this hypothesis, we examined activation of the PI3K-AKT-mTOR pathway in THP-1 macrophages infected with either WT or ΔMmpE strains. Immunoblot analysis demonstrated a time-dependent increase in the phosphorylation of AKT (Ser473), mTOR (Ser2448), and p70S6K (Thr389) following WT infection, whereas phosphorylation of these targets was attenuated in cells infected with the ΔMmpE mutant (*Figure 5—figure supplement 1C*). To investigate the roles of nuclear localization and phosphatase activity in MmpE-mediated signaling, THP-1 macrophages were infected with WT, ΔMmpE, Comp-MmpE, Comp-MmpE$^{\Delta NLS1}$, Comp-MmpE$^{\Delta NLS2}$, Comp-MmpE$^{\Delta NLS1-2}$, or Comp-MmpE-H348AN359A. Cells infected with WT or Comp-MmpE exhibited increased phosphorylation of Akt, mTOR, and p70S6K compared to those infected with ΔMmpE and Comp-MmpE$^{\Delta NLS1-2}$ (*Figure 5E and F*). These results further indicate that nuclear localization is the key determinant of MmpE-mediated activation of the PI3K-AKT-mTOR pathway during mycobacterial infection.

As PI3K-Akt-mTOR signaling is known to inhibit phagolysosomal fusion (*Yang et al., 2020*), and MmpE downregulates lysosomal genes (*Figure 4G*), we hypothesized that MmpE may impair lysosomal maturation. To test this, we examined intracellular trafficking of BCG in THP-1 macrophages infected with WT and the mutant strains. Confocal microscopy combined with LysoTracker staining was used to assess lysosomal acidification. We observed that ΔMmpE and MmpE$^{NLS1-2}$ mutants exhibited significantly higher co-localization with LysoTracker compared to WT and Comp-MmpE strains (*Figure 5G*), indicating that MmpE suppresses lysosomal maturation during infection. To determine whether this effect is mediated through the PI3K-AKT-mTOR pathway, THP-1 macrophages were treated with the dual PI3K/mTOR inhibitor BEZ235 at concentrations ranging from 0 to 200 nM (*Hsu et al., 2018*). Growth assays confirmed that BEZ235 had no effect on the viability of BCG strains over 8 days (*Figure 5—figure supplement 2A*). However, compared with WT, the ΔMmpE mutant showed reduced intracellular survival, and this difference was significantly attenuated upon BEZ235 treatment, suggesting that PI3K-AKT-mTOR signaling contributes to MmpE-mediated immune escape. We further examined the intracellular survival of WT MmpE, NLS-deleted mutant, and phosphatase-inactive strains in THP-1 macrophages treated with 100 nM BEZ235. Under untreated conditions, NLS-deficient and phosphatase-inactive strains showed significantly reduced intracellular survival relative to WT and Comp-MmpE, whereas BEZ235 treatment abolished these differences, providing further evidence that activation of the PI3K-AKT-mTOR pathway is essential for MmpE-mediated subversion of host antimicrobial responses during mycobacterial infection.

## MmpE suppresses the expression of *VDR* during BCG infection

Given the role of MmpE in modulating host signaling pathways, we next investigated whether it directly targets specific host genes during macrophage infection. Focusing on promoter regions (≤1 kb upstream of the transcription start site) of significantly upregulated genes (|Log$_2$FC >1| and $p$<0.05), we identified four candidate transcription factors, including *CREBBP*, *VDR*, *GREB1*, and *FOXP4* (*Figure 6A*). HOMER de novo motif analysis revealed a putative MmpE-binding motif specifically with the promoter of *VDR*, a key anti-inflammatory gene (*Aggeletopoulou et al., 2022*; *Figure 6B*). We next examined *VDR* expression in THP-1 macrophages infected with recombinant strains. Compared to the WT, *VDR* expression was significantly downregulated in cells infected with the ΔMmpE mutant

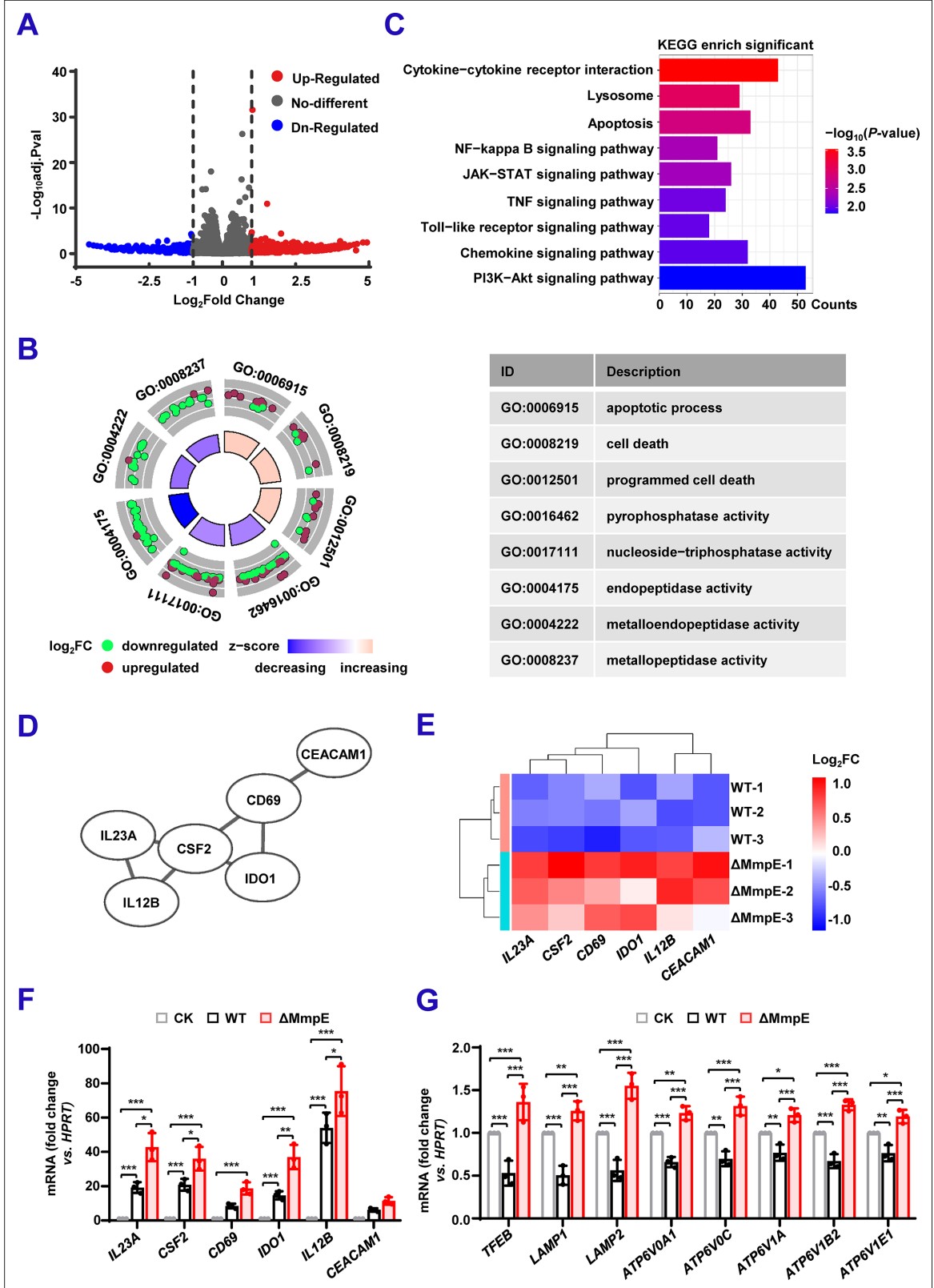

**Figure 4.** MmpE regulates host transcription network involved in inflammation response and lysosomal maturation. (**A**) Volcano plot showing differentially expressed genes (DEGs) in THP-1 cells infected with ΔMmpE versus wild-type (WT) strains. DEGs were defined as | log$_2$fold change |≥1 and $p<0.05$. (**B**) Enrichment analysis of DEGs shown in (**A**). The circular plot shows enriched biological process and molecular function terms. Outer rings display Gene ontology (GO) terms and associated genes, colored by expression change (upregulated, red; downregulated, green). The inner ring shows

*Figure 4 continued on next page*

*Figure 4 continued*

term z-scores (blue to pink). Selected terms are annotated with GO IDs and descriptions. (**C**) KEGG enrichment analysis of DEGs shown in (**A**). Bar length denotes the number of associated DEGs, and color represents statistical significance (-log₁₀(p-value)). (**D**) Protein-protein interaction network of immune-related DEGs, constructed using STRING v12.0 and visualized in Cytoscape. (**E**) Heatmap of immune-related DEGs in ΔMmpE-infected and WT-infected THP-1 cells. Log₂fold change values are shown across three biological replicates. Red and blue indicate upregulation and downregulation, respectively. (**F–G**) qRT-PCR validation of representative DEGs in THP-1 cells infected with WT or ΔMmpE for 24 hpi. Cytokine-related genes (**F**); lysosomal acidification and V-ATPase subunit genes (**G**). Data represent mean ± SD of three independent biological replicates. Statistical significance determined using two-way ANOVA, *$p<0.05$, **$p<0.01$, ***$p<0.001$.

The online version of this article includes the following figure supplement(s) for figure 4:

**Figure supplement 1.** MmpE modulates host transcription network involved in inflammation response and lysosomal maturation.

($Figure\ 6C$), suggesting that MmpE suppresses *VDR* transcription. Furthermore, we cloned the *VDR* promoter region (−100 bp to +50 bp) and conducted chromatin immunoprecipitation (ChIP) assays in HEK293T cells. While both *VDR* and *GAPDH* sequences were detected in input controls, *VDR* promoter DNA was selectively in ChIP samples from MmpE-EGFP-transfected cells (*Figure 6D and E*). In addition, the DNA-binding capacity of MmpE was unaffected by its phosphatase activity (*Figure 6F and G*). Electrophoretic mobility shift assays (EMSA) further confirmed the interaction, demonstrating a shifted band corresponding to the MmpE-VDR promoter complex and a dose-dependent increase in complex formation (*Figure 6H and I*). Collectively, these results suggest that MmpE binds the VDR promoter to modulate its transcription.

To further characterize the MmpE-regulated transcriptional network, we integrated CUT&Tag and RNA-seq data to identify 298 related genes (*Supplementary file 3*; *Figure 6—figure supplement 1A and B*). GO enrichment analysis revealed that these genes were significantly enriched in biological processes, such as cell differentiation and regulation of small GTPase-mediated signal transduction, both of which are fundamental to immune cell function, trafficking, and inflammatory responses (*Li et al., 2023*; *Ma et al., 2024*; *Figure 6—figure supplement 1C and D*). Collectively, these findings establish MmpE as a nucleomodulin that rewires host transcriptional programs involved in immune regulation and cellular defense.

## Nuclear translocation of MmpE is critical for mycobacterial survival in mice

To investigate the role of MmpE as a nucleomodulin in vivo, C57BL/6 mice were infected with WT, ΔMmpE, Comp-MmpE, and Comp-MmpE^ΔNLS1-2 strains. Compared with mice infected with WT or Comp-MmpE strains, mice infected with ΔMmpE or Comp-MmpE^ΔNLS1-2 strains had significantly lower bacterial loads in their lungs (*Figure 7A*). Similar results were observed in splenic bacterial colonization (*Figure 7—figure supplement 1A*). Hematoxylin and eosin (H&E) staining of lung tissues from the mice infected with WT or Comp-MmpE strains showed extensive inflammatory infiltrates, while the lungs of the mice infected with ΔMmpE or Comp-MmpE^ΔNLS1-2 strains exhibited markedly reduced inflammation (*Figure 7B*). In addition, compared with mice infected with WT or Comp-MmpE strains, mice infected with ΔMmpE or Comp-MmpE^ΔNLS1-2 strains showed a significant increase in *IL1a*, *IL1b*, and *IL6* in the spleen (*Figure 7C and E*). These results indicate that the nuclear translocation of MmpE is critical for mycobacterial intracellular survival.

Since MmpE modulates host transcriptional responses in THP-1 macrophages, we examined the expression of *Vdr*, inflammation-related, and lysosome-associated genes in the lungs and spleens of mice at 14 and 28 days post infection (dpi). *Vdr* expression was significantly reduced in both lungs and spleens infected with the ΔMmpE or Comp-MmpE^ΔNLS1-2 strains compared to those infected with WT or Comp-*MmpE* strains (*Figure 7—figure supplement 1B*). Consistent with Hematoxylin and eosin observation, the expression of inflammation-related genes (e.g. *Il23a*, *Csf2*, *Cd69*, *Ido1*, *Il12b*, and *Ceacam1*) were markedly decreased in lungs infected with the ΔMmpE or Comp-MmpE^ΔNLS1-2, but significantly elevated in spleens from the same groups (*Figure 7—figure supplement 1C–D*). Moreover, as observed in vitro, lysosomal maturation-associated genes, such as *Tfeb*, *Lamp1*, *Lamp2*, together with genes encoding V-ATPase subunits, were upregulated in both lung and spleen tissues of the mice infected with ΔMmpE or Comp-MmpE^ΔNLS1-2 compared to control mice (*Figure 7— figure supplement 1E–H*). Taken together, our experiments demonstrate that MmpE promotes

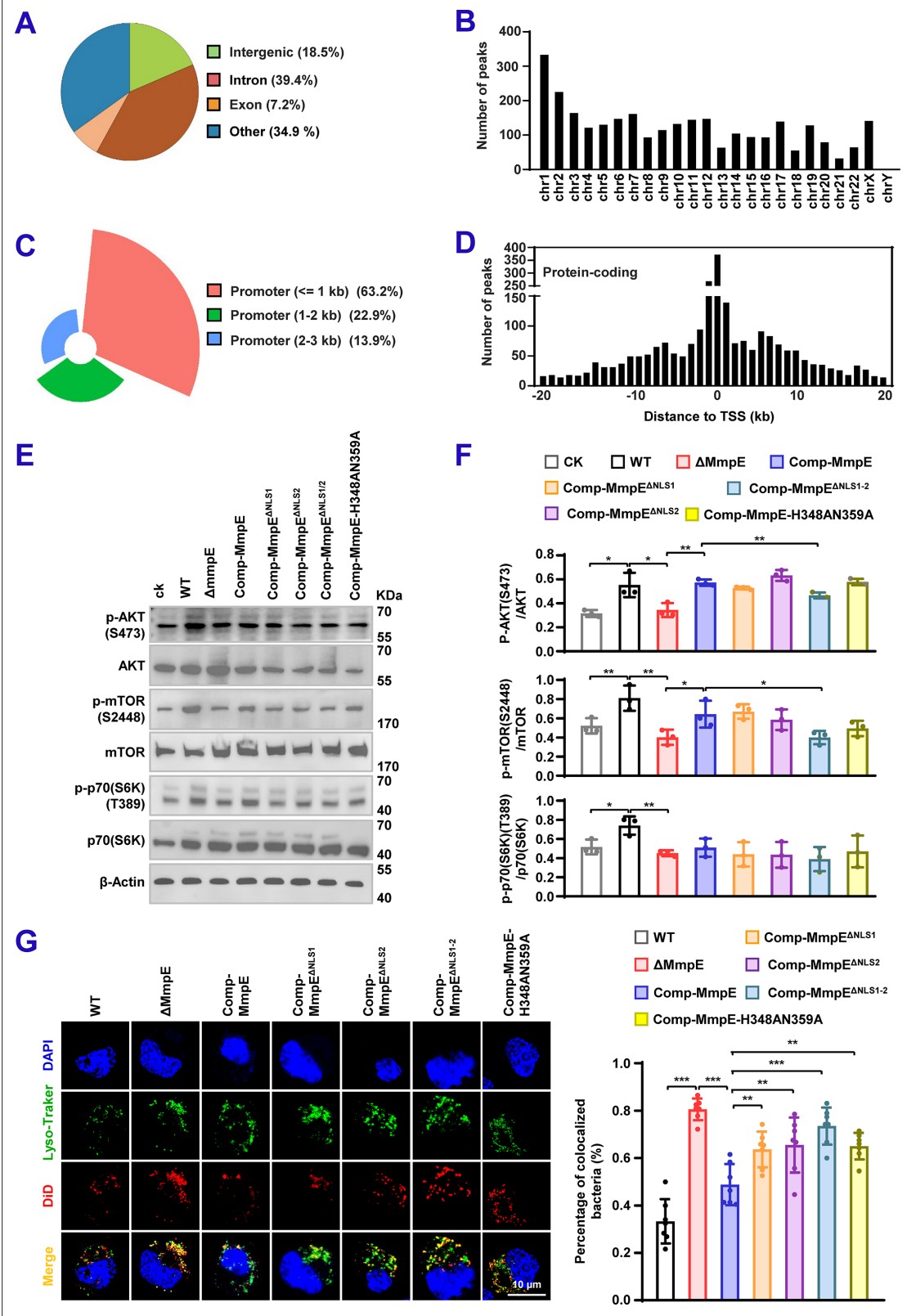

**Figure 5.** MmpE regulates the PI3K-Akt-mTOR signaling pathway during macrophage infection. (**A–D**) Cut&Tag analysis of MmpE-binding regions in HEK293T cells. (**A**) Genomic distribution of potential MmpE-binding regions in HEK293T cells. (**B**) Chromosomal localization of MmpE-enriched peaks (fold enrichment > 9). (**C**) Biotype distribution of potential MmpE-binding regions in HEK293T cells. (**D**) Distribution of binding sites relative to the nearest transcription start sites (TSS) within ± 20 kb of protein-coding genes. (**E**) Immunoblot analysis of pathway activation in THP-1 macrophages

*Figure 5 continued on next page*

*Figure 5 continued*

infected with wild-type (WT), ΔMmpE, Comp-MmpE, Comp-MmpE$^{ΔNLS1}$, Comp-MmpE$^{ΔNLS2}$, Comp-MmpE$^{ΔNLS1-2}$, or Comp-MmpE-H348AN359A. Phosphorylation of Akt, mTOR, and p70S6K was evaluated at 8 hpi. (**F**) Quantification of phosphorylation levels across time points. (**G**) (left) Confocal microscopy analysis of phagosome acidification in infected THP-1 macrophages at 24 hpi. Cells were stained with LysoTracker Red, and BCG strains were stained with DiD. (Right) Co-localization between mycobacteria and LysoTracker was assessed. Representative images showing co-localization of intracellular bacteria with LysoTracker signal. Scale bar, 10 μm. Data represent mean ± SD of three independent biological replicates. Statistical significance determined using two-way ANOVA or two-tailed unpaired Student's *t*-tests, *$p<0.05$, **$p<0.01$, ***$p<0.001$.

The online version of this article includes the following source data and figure supplement(s) for figure 5:

**Source data 1.** Original western blots for panel E, indicating the relevant bands.

**Source data 2.** Original files for western blot analysis displayed in panel E.

**Figure supplement 1.** MmpE regulates the PI3K-Akt-mTOR signaling pathway during macrophage infection.

**Figure supplement 1—source data 1.** Original western blots for panel C, indicating the relevant bands.

**Figure supplement 1—source data 2.** Original files for western blot analysis displayed in panel C.

**Figure supplement 2.** MmpE promotes pathogen intracellular survival via the PI3K-Akt-mTOR signaling pathway.

(**A–C**) Effects of BEZ235 on the growth and intracellular survival of BCG strains. Bacterial growth in 7H9 medium with or without BEZ235 (0–200 nM) (A). Intracellular survival of wild-type (WT) and ΔMmpE strains in THP-1 macrophages treated with or without BEZ235 (0–200 nM); CFUs were quantified at 48 hpi (B). Intracellular survival of WT, ΔMmpE, Comp-MmpE, Comp-MmpE$^{ΔNLS1}$, Comp-MmpE$^{ΔNLS2}$, Comp-MmpE$^{ΔNLS1-2}$, Comp-MmpE-H348AN359A in THP-1 macrophages treated with or without 100 nM BEZ235; CFUs were determined at 4, 48 hpi (C). Data represent mean ± SD of three independent biological replicates. Statistical significance determined using two-way ANOVA, *$p<0.05$, **$p<0.01$, ***$p<0.001$.

mycobacterial survival in vivo through its nuclear translocation, likely by modulating host immune responses and lysosomal maturation pathways.

## Discussion

### MmpE is a bifunctional virulence factor in Mtb

Previously, *Forrellad et al., 2020* characterized Rv2577/MmpE of Mtb as an alkaline phosphatase/phosphodiesterase that affects phagosome maturation arrest and virulence of Mtb. While the enzymatic activity and virulence-related phenotype of MmpE have been established, specific cofactor dependencies, catalytic residues, and the mechanisms underlying host-pathogen interactions remain inadequately defined. Here, our study shows that MmpE possesses $Fe^{3+}$-dependent phosphatase activity, with H348 and N359 identified as critical catalytic residues (*Figure 2C–E*). Moreover, deletion of these residues significantly impairs the survival of recombinant BCG strains in THP-1 macrophages, indicating that its phosphatase activity is essential for intracellular persistence (*Figure 3E*). Furthermore, our study identifies MmpE as a novel nucleomodulin that translocates into the host nucleus primarily via its nuclear localization signal (NLS; RRK$^{460-462}$) (*Figure 1D-G*, *Figure 1—figure supplement 1D*). Upon infection, MmpE modulates a broad range of host immune signaling pathways, including cytokine-cytokine receptor interactions, TNF, Toll-like receptor, chemokine, NF-κB, JAK-STAT, and PI3K-Akt pathways. Specifically, MmpE enhances the phosphorylation levels of AKT (Ser473), mTOR (Ser2448), and p70S6K (Thr389), modulating the PI3K-AKT-mTOR signaling pathway and ultimately impairing lysosomal maturation to promote intracellular survival of the pathogen (*Figure 5E and G*, *Figure 5—figure supplement 1C*; *Figure 5—figure supplement 2B and C*). Additionally, it also regulates the expression of key transcription factors, such as VDR, CREBBP, GREB1, and FOXP4 (*Figure 6A*). Notably, the deletion of the NLS significantly impairs the functionality of MmpE, thereby reducing the survival of BCG strains in both macrophages and murine models (*Figures 3A–D and 7A*). Taken together, these findings indicate that MmpE functions as a bifunctional virulence factor, combining phosphatase activity with nuclear localization to modulate host responses during infection.

### Targeting host *VDR* gene might represent a unique immune evasion strategy by Mtb

Nucleomodulins are effector proteins secreted by bacterial pathogens that specifically target the host cell nucleus, where they modulate nuclear processes to subvert host responses (*Hanford et al., 2021*). Their mechanisms of action within the nucleus typically involve indirect interference with host

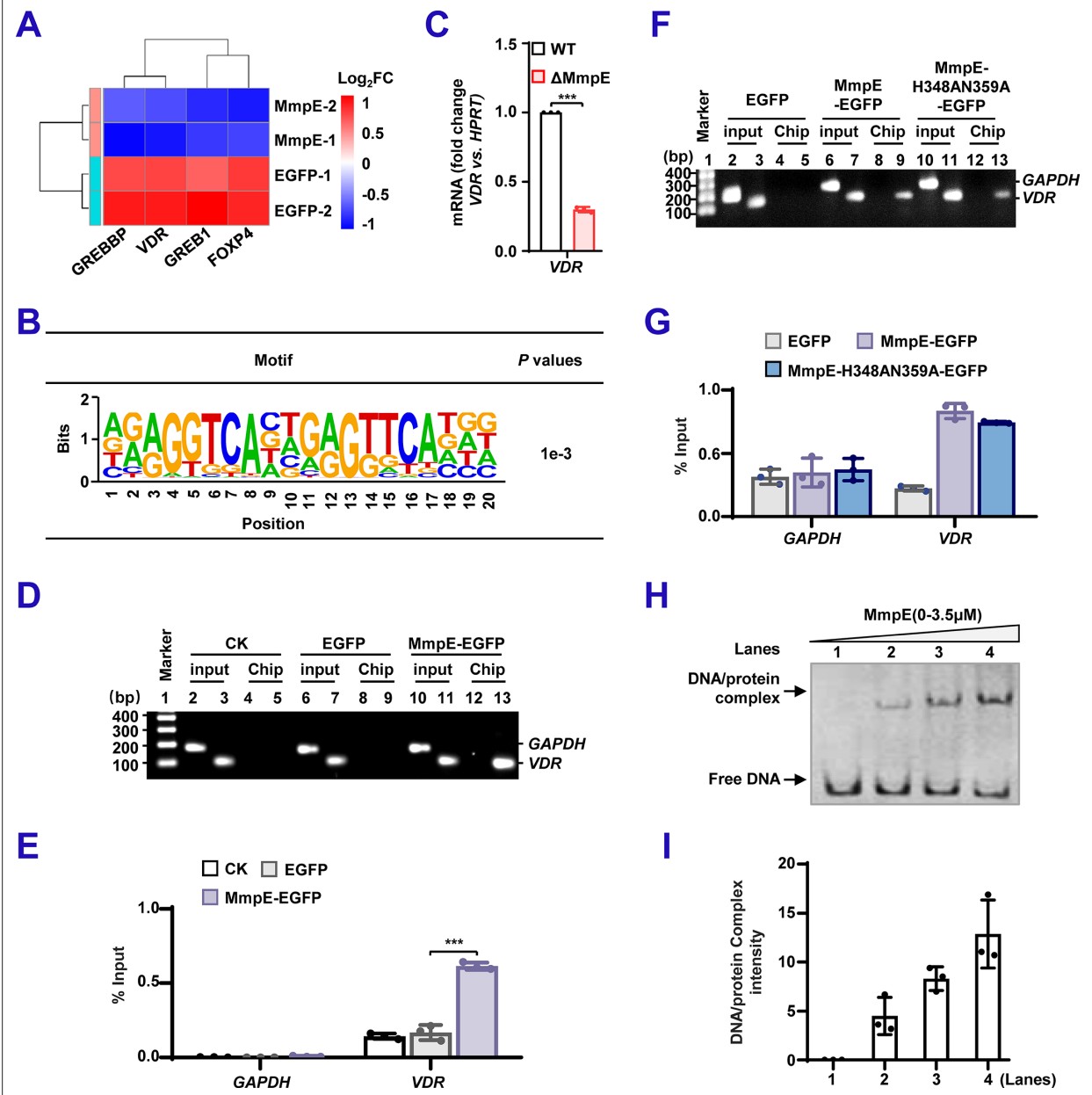

**Figure 6.** MmpE suppresses the expression of vitamin D receptor (*VDR*) during BCG infection. (**A**) Heatmap of transcription factors associated with MmpE ChIP-seq peaks in HEK293T cells. Color scale indicates expression changes (red: upregulated; blue: downregulated). (**B**) De novo motif analysis of MmpE-bound sequences using HOMER. Enrichment *p*-values were calculated with TOMTOM. '% of targets' indicates the proportion of peaks containing each motif; letter height reflects nucleotide frequency. (**C**) Quantitative RT-PCR analysis of *VDR* expression in THP-1 cells infected with wild-type (WT) or ΔMmpE strains for 24 hr. (**D–G**) ChIP-PCR and qPCR analysis of the *VDR* promoter region. HEK293T cells were transfected with EGFP, MmpE-EGFP, or MmpE-H348AN359A-EGFP. Chromatin was immunoprecipitated using anti-GFP antibody. PCR was performed with primers targeting the *VDR* promoter and GAPDH (negative control), and products were analyzed by agarose electrophoresis (**D, F**). Enrichment was quantified by qPCR using the $2^{-\Delta Ct}$ method (**E, G**). (**H–I**) EMSA using purified MmpE protein and a DNA probe corresponding to the *VDR* promoter. Samples were resolved by native PAGE. Arrows indicate the positions of free DNA and slower-migrating species (**H**); signal intensities were quantified by densitometry (**I**). Data represent mean ± SD of three independent biological replicates. Statistical significance determined using two-way ANOVA or two-tailed unpaired Student's *t*-tests, ***p<0.001.

The online version of this article includes the following source data and figure supplement(s) for figure 6:

**Source data 1.** Original agarose gel images for panels D and F and EMSA images for panel H, indicating the relevant bands.

**Source data 2.** Original files for agarose gel electrophoresis displayed in panels D and F, and for EMSA displayed in panel H.

**Figure supplement 1.** MmpE modulates the transcription of immune-associated genes.

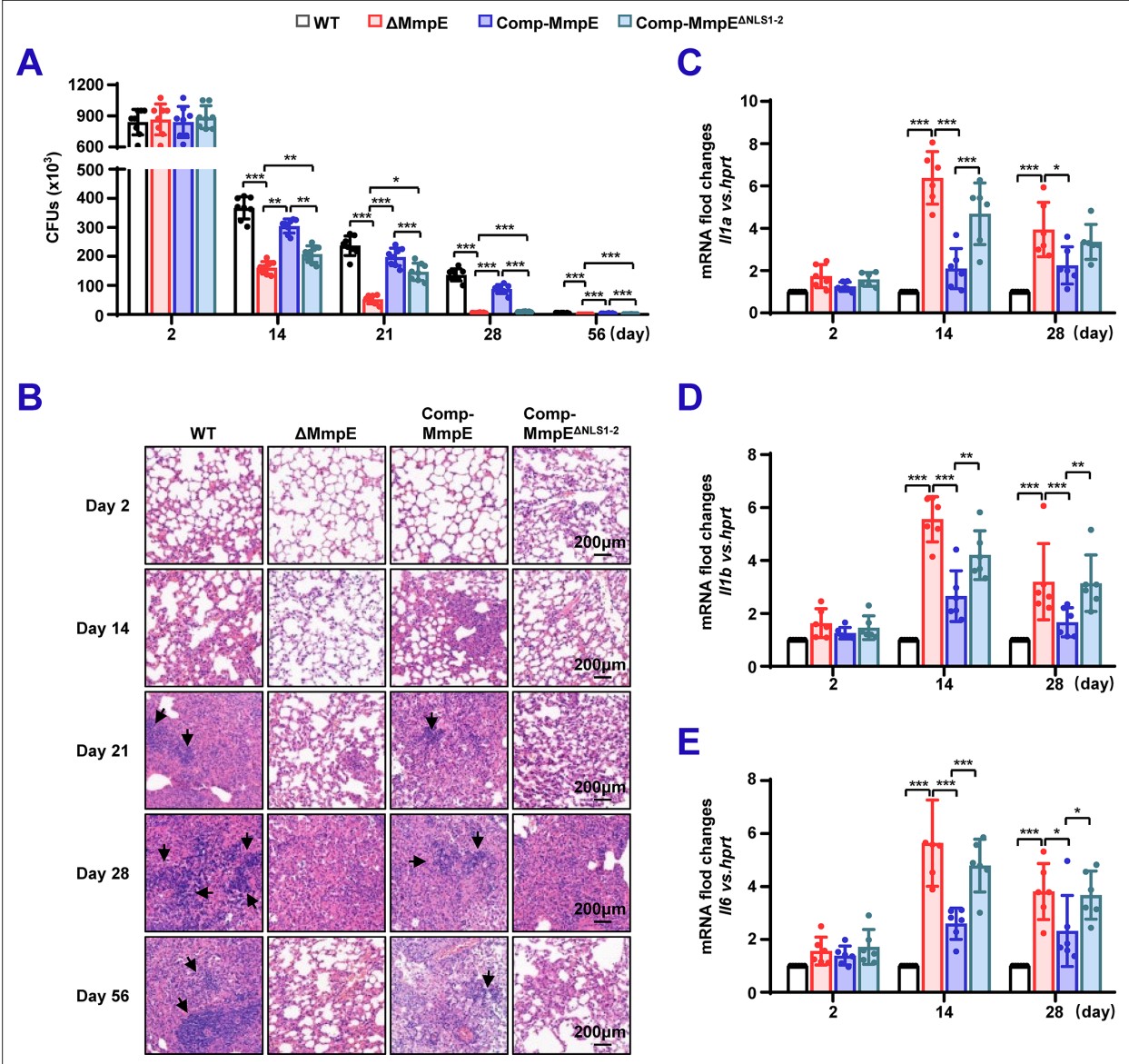

**Figure 7.** Nuclear translocation of MmpE is essential for mycobacterial survival in mice. (**A**) Bacterial burden in lungs of infected mice. Specific pathogen-free (SPF) C57BL/6 mice (n=6 per group) were intranasally infected with $1.0×10^7$ colony-forming units (CFUs) of BCG strains, including wild-type (WT), ΔMmpE, Comp-MmpE, or Comp-MmpE$^{ΔNLS1-2}$. Bacterial titers in lung homogenates were quantified by CFU assays at 0, 14, 28, and 56 dpi. (**B**) Histopathology of infected lung tissues. Hematoxylin and eosin (H&E)-stained lung sections from mice infected as in (**A**) show granulomatous inflammation. Scale bars: 200 μm. (**C–D**) Pro-inflammatory cytokine expression in the spleen of infected mice. qRT-PCR analysis of cytokine mRNA levels of *Il1a* (**C**), *Il1b* (**D**) and *Il6* (**E**) in spleen tissues from infected mice (n=6/group) at 2–28 dpi. Data represent mean ± SD of three independent biological replicates. Statistical significance determined using two-way ANOVA, *$p<0.05$, **$p<0.01$, ***$p<0.001$.

The online version of this article includes the following figure supplement(s) for figure 7:

**Figure supplement 1.** MmpE facilitates bacterial colonization in the spleens of infected mice.

transcriptional programs, such as through epigenetic modifications, chromatin remodeling, or inhibition of key signaling pathways (*Jose et al., 2016*; *Singh et al., 2023*). Despite extensive characterization of nucleomodulin functions, direct binding to host gene promoters remains rare and mechanistically distinct. In this study, we found that MmpE directly binds to the promoter of the *VDR* gene (*Figure 6*) and attenuates the expression of VDR-regulated inflammatory genes (*Zhou et al., 2020*; *Yang et al., 2019*; *Figure 3F–H*). VDR is a critical nuclear receptor that controls the expression of numerous genes involved in host immune defense, including antimicrobial peptides and cytokines (*Usategui-Martín et al., 2022*). For example, Mtb suppresses the expression of the host

gene *CYP27B1* to reduce active vitamin D synthesis, thereby limiting VDR activation and downstream antimicrobial responses (*Liu et al., 2006*). Similarly, *Salmonella* downregulates *VDR* through miRNA-mediated mechanisms, weakening mucosal immunity (*Chen et al., 2014*). This mechanism of MmpE regulating VDR expression reveals a novel strategy by bacterial pathogens to evade host defenses through direct manipulation of host gene promoters by nucleomodulins.

## MmpE regulates the PI3K-Akt-mTOR pathway to inhibit lysosomal maturation and enhance pathogen survival

Mtb employs multiple strategies to evade phagolysosomal degradation, a process essential for bacterial clearance (*Chandra et al., 2022*; *Zhang et al., 2023*; *Singh and Nagaraja, 2025*). For example, approximately 70% of phagosomes harboring Mtb fail to fuse with lysosomes, partly due to the activity of secreted phosphatases, such as PtpA, PknG, and SapM, which disrupt vesicular trafficking (*Wong et al., 2011*; *Ge et al., 2022*; *Zhang et al., 2024*; *Alsayed and Gunosewoyo, 2024*). MmpE has also been shown to contribute to the arrest of lysosomal maturation in human cells; however, the underlying mechanisms remain unclear (*Forrellad et al., 2020*).

In this study, we elucidate a mechanism by which MmpE inhibits lysosomal maturation. The absence of MmpE during BCG infection resulted in substantial upregulation of lysosome-associated genes, including *TFEB*, a master regulator of lysosomal biogenesis and function, underscoring MmpE's role in lysosomal regulation (*Jeong et al., 2021*; *Wang et al., 2024*; *Figure 4G*, *Figure 4—figure supplement 1C*). Moreover, the ΔMmpE and MmpE$^{\Delta NLS1-2}$ mutants exhibited significantly increased expression of lysosomal maturation-associated genes in both mice and infected THP-1 cells, along with enhanced LysoTracker co-localization in THP-1 cells, compared to WT and Comp-MmpE strains (*Figure 4G*, *Figure 5G*, *Figure 7—figure supplement 1C–F*). Collectively, these findings indicate that MmpE inhibits lysosomal maturation during mycobacterial infection. Furthermore, CUT&Tag analysis revealed that MmpE binds to the promoter regions of host genes within the PI3K-Akt-mTOR signaling pathway, including *PRKCB*, *PLCG2*, and *PIK3CB*, which are key upstream regulators of TFEB activity (*Fang et al., 2021*; *Gounis et al., 2025*; *Figure 5—figure supplement 1A*). Additionally, during infection, WT strains exhibited significantly elevated phosphorylation of Akt, mTOR, and p70S6K compared to the ΔMmpE and the Comp-MmpE$^{\Delta NLS1-2}$ strain (*Figure 5E and F*, *Figure 5—figure supplement 1C*). When THP-1 macrophages were treated with the dual PI3K/mTOR inhibitor BEZ235, the survival difference between WT and MmpE mutants was significantly reduced. These results collectively indicate that MmpE regulates the PI3K-Akt-mTOR-TFEB axis, thereby inhibiting lysosomal maturation and enhancing bacterial survival in the host.

In summary, our study identifies MmpE as a novel bifunctional virulence factor of mycobacteria, combining Fe$^{3+}$-dependent phosphatase activity with nuclear translocation capability. MmpE translocates into the host nucleus and directly binds to the promoters of some key host genes, such as *VDR*, thereby inhibiting the expression of the downstream targets implicated in the inflammation responses and lysosomal maturation, thereby promoting mycobacterial survival in macrophages and in mice (*Figure 8*). These findings reveal a sophisticated mechanism by which bacterial pathogens manipulate host gene transcription and signaling pathways to promote intracellular persistence.

## Materials and methods

### Bacterial strains and culture conditions

The plasmids and bacterial strains used in this study are detailed in *Supplementary files 4 and 5*, respectively. *E. coli* DH5α and BL21 were cultured in Luria-Bertani (LB) medium under standard conditions. *M. bovis* BCG strains were grown in Middlebrook 7H9 broth supplemented with 10% (v/v) OADC (oleic acid, albumin, dextrose, catalase), 0.05% (v/v) Tween-80, and 2% (v/v) glycerol. Antibiotics were added as required at the following final concentrations: 50 µg/mL hygromycin (for mycobacteria) and 50 µg/mL kanamycin (for both mycobacteria and *E. coli*).

### Cell culture

THP-1 cells were cultured in Roswell Park Memorial Institute 1640 (RPMI 1640) medium supplemented with 10% (v/v) heat-inactivated fetal bovine serum (FBS), 1 mM sodium pyruvate, 2 mM L-glutamine, 10 mM HEPES buffer (pH 7.2–7.5), and 50 µM 2-mercaptoethanol. For differentiation into

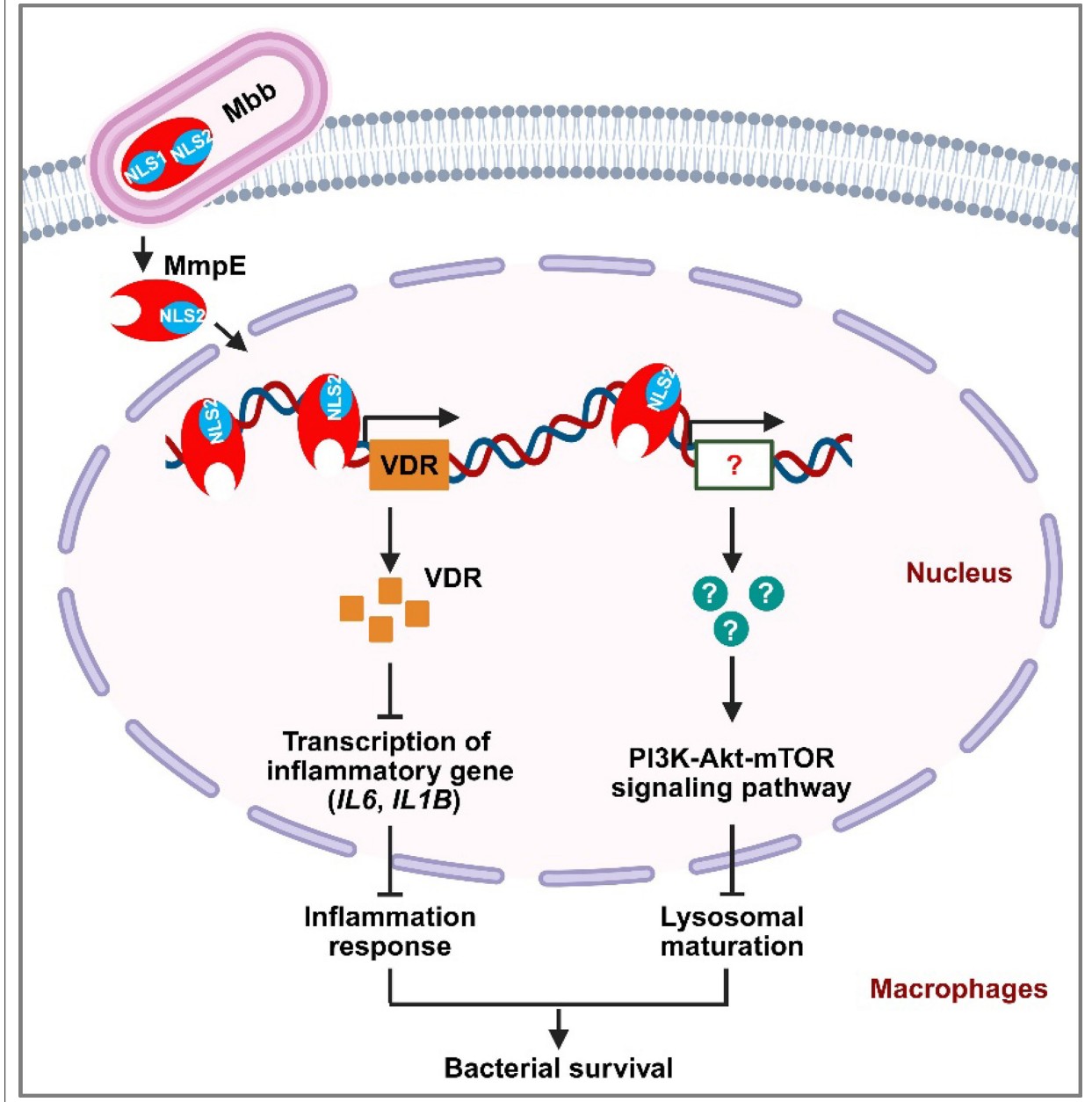

**Figure 8.** Schematic diagram of nucleomodulin MmpE-mediated immune suppression and enhanced mycobacterial survival. During infection, the nucleomodulin MmpE translocates into the host nucleus via its C-terminal NLS2 motif (RRK$^{460-462}$), where it binds to the vitamin D receptor (*VDR*) promoter and suppresses transcription of inflammatory genes. Simultaneously, MmpE regulates the PI3K-Akt-mTOR signaling pathway, thereby inhibiting lysosome maturation. These dual mechanisms contribute to immune evasion and promote intracellular survival of mycobacteria.

macrophages, THP-1 cells were treated with 200 nM phorbol 12-myristate 13-acetate (PMA) for 24 hr to allow complete differentiation prior to experimental use.

HEK293T cells and RAW264.7 macrophages were cultured in Dulbecco's modified Eagle's medium (DMEM) supplemented with 10% (v/v) FBS and 50 µg/mL penicillin-streptomycin. All cells were maintained at 37°C in a humidified atmosphere with 5% $CO_2$.

### Animal experiments

Specific pathogen-free (SPF) female C57BL/6 mice (6–8 weeks old, 16–18 g) were obtained from Chang-sheng Bio (Liaoning, China). All animals were maintained under SPF conditions in individually

ventilated cages with controlled temperature and humidity, on a 12 hr light/dark cycle. Mice had ad libitum access to sterilized food and water.

## Phylogenetic and sequence analysis

Homologs of the metallophosphatase MmpE were identified from 11 *Mycobacterium* species, including: *M. tuberculosis* (Mtu), *M. bovis* BCG (Mbb), *M. orygis* (Mory), *M. kubicae* (Mku), *M. paraterrae* (Mpaa), *M. farcinogenes* (Mfg), *M. mucogenicum* (Mmuc), *M. vicinigordonae* (Mgor), *M. lentiflavum* (Mlw), *M. avium* (Mav), *M. manitobense* (Mmam). Genomic sequences were retrieved from the NCBI RefSeq database and filtered for completeness.

Protein sequences were aligned using Clustal Omega v1.2.4 with default parameters (gap opening penalty = 10, gap extension = 0.2, and iterative refinement enabled). Conserved residues (≥90% identity) were visualized using ESPript 3.0. Phylogenetic reconstruction was performed in MEGA v12.0 using the neighbor-joining algorithm with pairwise deletion for gap treatment and a Poisson substitution model, appropriate for closely related bacterial sequences. Bootstrap support was calculated with 1,000 replicates, and nodes with values ≥50% were retained. The resulting tree was visualized using the iTOL platform, without additional pruning.

## Structural analysis of MmpE

Structural models of full-length MmpE and its NLS-deletion mutants (MmpE$^{\Delta NLS1}$, MmpE$^{\Delta NLS2}$, MmpE$^{\Delta NLS1-2}$) were generated using AlphaFold v2.2.0. Predictions were performed with default settings, including the use of the pre-trained model and MSA generation via UniRef90 and MGnify. Model confidence was assessed using pLDDT scores. All structures were analyzed in UCSF ChimeraX for conformational differences between mutants. The AlphaFold-predicted reference model was further examined in PyMOL to identify putative metal-binding residues based on spatial proximity and coordination geometry within the predicted active site.

## Cell transfection and confocal microscopy

HEK293T cells were seeded at 70% confluency in poly-L-lysine-coated 35 mm glass-bottom dishes and maintained in DMEM supplemented with 10% (v/v) FBS at 37°C with 5% $CO_2$. Transfection was performed using the Hieff Trans Liposomal Transfection Reagent according to the manufacturer's protocol. Briefly, 0.5 μg of plasmid DNA and 1.5 μL of transfection reagent were diluted in 250 μL of Opti-MEM, incubated at room temperature for 20 min, and added dropwise to each well.

At 24–48 hpt, cells were washed with PBS, fixed in 4% paraformaldehyde for 15 min, and permeabilized with 0.1% Triton X-100 for 10 min. Nuclei were counterstained with 1 μg/mL DAPI for 10 min. Imaging was performed using an Olympus FV1000 Confocal Laser Scanning Microscope equipped with a 100x/1.40 NA oil immersion objective. Fluorescence was detected using the following parameters: EGFP, excitation at 488 nm, emission at 509 nm; DAPI, excitation at 405 nm, emission at 461 nm. Spectral unmixing was applied using FV10-ASW v4.2 to minimize channel cross-talk. Z-stack images were acquired in sequential mode with a step size of 0.5 μm to ensure consistency across samples.

## Cell fractionation and immunoblotting

Cells were lysed in RIPA buffer supplemented with 1% (v/v) protease inhibitor cocktail, and protein concentrations were determined using the BCA assay. Equal amounts of protein were separated by SDS-PAGE and transferred onto PVDF membranes. Membranes were blocked for 1 hr at room temperature with 5% (w/v) non-fat milk in TBST (25 mM Tris, 150 mM NaCl, 2 mM KCl, 0.2% Tween-20, pH 7.4), then incubated overnight at 4°C with primary antibodies. After three 6 min washes with TBST, membranes were incubated with HRP-conjugated secondary antibodies for 1 h at room temperature. Signal detection was performed using enhanced chemiluminescence (ECL), and images were acquired with the ChemiDoc MP Imaging System (Bio-Rad).

For subcellular fractionation, nuclear and cytoplasmic protein fractions were prepared using the Nuclear and Cytoplasmic Protein Extraction Kit (Beyotime) with minor modifications. Cells were washed with PBS, detached using a cell scraper, and pelleted by centrifugation at 2000× *g* for 5 min. The pellet (~20 μL packed cells) was resuspended in 150 μL of Cytoplasmic Extraction Reagent A (with PMSF), vortexed for 5 s, and incubated on ice for 15 min. After the addition of 10 μL Reagent B, the sample was vortexed briefly, incubated on ice, and centrifuged at 12,000× *g* for 5 min at 4°C

to collect the cytoplasmic fraction (supernatant). The remaining pellet was extracted with 50 μL of Nuclear Extraction Reagent (with PMSF), vortexed intermittently for 30 min on ice, and centrifuged at 12,000× $g$ for 10 min at 4°C to obtain the nuclear fraction. β-actin, or histone H3 was used as a loading control for cytoplasmic or nuclear fractions. Antibody information is listed in the Key resources table.

**Key resources table**

| Reagent type (species) or resource | Designation | Source or reference | Identifiers | Additional information |
|---|---|---|---|---|
| Strain, strain background (*Mycobacterium bovis* BCG) | *Mycobacterium bovis* BCG | ATCC | Cat#35734 | |
| Strain, strain background (*Escherichia coli*) | ArcticExpress(DE3)pRARE2 | ANGYUBIO | Cat#G6023-2 | |
| Cell line (Homo-sapiens) | HEK293T | Cellosaurus | CVCL_0063 | |
| Cell line (Homo-sapiens) | THP-1 | Cellosaurus | CVCL_0006 | |
| Cell line (*Mus musculus*) | RAW264.7 | Cellosaurus | CVCL_C6XG | |
| Antibody | Anti-β-actin (Mouse Monoclonal Antibody) | Abbkine | Cat#ABL1010 | WB 1:10000 |
| Antibody | Anti-Histone H3 (Mouse Monoclonal Antibody) | Abbkine | Cat#ABL1070 | WB 1:2000 |
| Antibody | Goat anti-rabbit IgG | Abbkine | Cat#A21020 | WB 1:10000 |
| Antibody | Goat anti-mouse IgG | Abbkine | Cat#A25012 | WB 1:1000 |
| Antibody | GFP tag (Mouse Monoclonal antibody) | Proteintech | RRID:AB_11182611 | WB 1:10000 |
| Antibody | Flag tag (Mouse Monoclonal antibody) | Proteintech | RRID:AB_2918475 | WB 1:10000 |
| Antibody | AKT (Rabbit Polyclonal antibody) | Proteintech | RRID:AB_2224574 | WB 1:5000 |
| Antibody | Phospho-AKT (Ser473) (Mouse Monoclonal antibody) | Proteintech | RRID:AB_2782958 | WB 1:5000 |
| Antibody | Mtor (Mouse Monoclonal antibody) | Proteintech | RRID:AB_2882219 | WB 1:10000 |
| Antibody | Phospho-mTOR (Ser2448) (Mouse Monoclonal antibody) | Proteintech | RRID:AB_2889842 | WB 1:10000 |
| Antibody | p70/S6K (Rabbit Polyclonal antibody) | Proteintech | RRID:AB_2269787 | WB 1:10000 |
| Antibody | p-p70(S6K)(T389) (Rabbit monoclonal antibody) | Proteintech | RRID:AB_3086477 | WB 1:10000 |
| Antibody | Ag85B | This paper | N/A | |
| Antibody | GlpX | This paper | N/A | |
| Chemical compound | DMEM Medium | Gibco | Cat#11965092 | |
| Chemical compound | RPMI 1640 Medium | Gibco | Cat#C11875500BT | |
| Chemical compound | Opti-MEM Medium | Gibco | Cat#31985070 | |
| Chemical compound | FBS | Gibco | Cat#A5256701 | |
| Chemical compound | Sodium pyruvate | Gibco | Cat#11360070 | |
| Chemical compound | L-glutamine | Gibco | Cat#A2916801 | |
| Chemical compound | HEPES | Gibco | Cat#15630080 | |
| Chemical compound | 2-Mercaptoethanol | Gibco | Cat#21985023 | |
| Chemical compound | penicillin-streptomycin antibiotics | Gibco | Cat#15140122 | |
| Chemical compound | PMA | Sigma-Aldrich | Cat#P8139 | |
| Chemical compound | p-NPP | Sigma-Aldrich | Cat#4264-83-9 | |
| Chemical compound | Formaldehyde | Sigma-Aldrich | Cat#252549 | |
| Chemical compound | OADC | MilliporeSigma | Cat#M0678 | |

*Continued on next page*

*Continued*

| Reagent type (species) or resource | Designation | Source or reference | Identifiers | Additional information |
|---|---|---|---|---|
| Chemical compound | DAPI | Beyotime | Cat# P0126 | |
| Chemical compound | Triton-X-100 | Beyotime | Cat#P0096 | |
| Chemical compound | RIPA buffer | Beyotime | Cat#P0038 | |
| Chemical compound | protease inhibitor cocktail | Boster | Cat#AR1182 | |
| Chemical compound | HieffTrans Liposomal Transfection Reagent | YEASEN | Cat#40802ES03 | |
| Chemical compound | protein A/G magnetic beads | MedChemExpress | Cat#HY-K0202 | |
| Commercial assay or kit | TRIpure Reagent | Aidlab | Cat#RN0101 | |
| Commercial assay or kit | EASYspin RNA Mini Kit | Aidlab | Cat#RN0702 | |
| Commercial assay or kit | EndoFreePlasmidMiniKit | Aidlab | Cat#PL0401 | |
| Commercial assay or kit | cDNA Reverse Transcription Kit | Vazyme | Cat#R333-01 | |
| Commercial assay or kit | 2×ChamQ Universal SYBR qPCR Master Mix | Vazyme | Cat#Q711-03 | |
| Commercial assay or kit | Uniclone One Step Seamless Cloning Kit | Genesand | Cat#SC612 | |
| Commercial assay or kit | Clarity Western ECL Substrate | BIO-RAD | Cat#1705060 | |
| Commercial assay or kit | Nuclear and Cytoplasmic Protein Extraction kit | Beyotime | Cat#P0027 | |
| Software | AlphaFold v2.2.0 | SciCrunch Registry | RRID:SCR_023662 | |
| Software | GraphPad Prism 8 | SciCrunch Registry | RRID:SCR_002798 | |
| Software | UCSF Chimera | SciCrunch Registry | RRID:SCR_004097 | |
| Software | MEGA 12.0 | SciCrunch Registry | RRID:SCR_000667 | |
| Software | Cytoscape 3.10.3 | SciCrunch Registry | RRID:SCR_003032 | |
| Software/Viewers | CaseViewer v2.0 | SciCrunch Registry | RRID:SCR_017654 | |
| Software, algorithm | jvenn / Venny 2.1.0 | SciCrunch Registry | RRID:SCR_016343 | |
| Software, algorithm | iTOL | SciCrunch Registry | RRID:SCR_018174 | |
| Software, algorithm | ESPript | SciCrunch Registry | RRID:SCR_006587 | |
| Software, algorithm | SignalP 5.0 | SciCrunch Registry | RRID:SCR_015644 | |
| Software, algorithm | STRING 12.0 | SciCrunch Registry | RRID:SCR_005223 | |

## Immunoblot analysis of MmpE secretion

The procedure for immunoblot analysis of MmpE secretion was adapted from previous studies with minor modifications (*Zhang et al., 2022*; *Chen et al., 2025*). BCG strains expressing C-terminally Flag-tagged MmpE were cultured in Middlebrook 7H9 broth with 0.5% glycerol, 0.02% Tyloxapol, and 50 μg/mL kanamycin at 37°C (80 rpm) until OD600 ~0.6. Cells were harvested by centrifugation (4000×$g$, 15 min, 4°C), and supernatants filtered through 0.22 μm PES membranes. Filtrates were concentrated ~50 fold using Amicon Ultra-15 filters (10 kDa cutoff) for secreted MmpE detection. Bacterial pellets were washed, resuspended in lysis buffer (50 mM Tris-HCl, pH 7.5; 150 mM NaCl; 1 mM PMSF; protease inhibitors). Lysates were clarified by centrifugation (10,000×$g$, 15 min, 4°C). Whole-cell lysates and concentrated culture filtrates were analyzed by immunoblotting as described above, using anti-Flag antibodies for detection.

## Protein expression and purification

N-terminal maltose-binding protein (MBP)-tagged and C-terminal His$_6$-tagged wild-type MmpE, as well as mutant lacking the Tat signal peptide were expressed in *E. coli* BL21 (DE3). Protein expression was performed in 1 L of LB medium. Cultures were grown at 37°C until the optical density at 600 nm

(OD$_{600}$) reached approximately 0.8, after which protein expression was induced with 0.2 mM IPTG at 18°C for 24 hr. Following induction, cells were harvested by centrifugation and lysed by sonication in ice-cold lysis buffer containing 150 mM NaCl, 30 mM Tris-HCl (pH 7.5), and 1 mM PMSF. The lysate was clarified by centrifugation, and the supernatant was subjected to affinity purification using an MBP affinity column.

## Phosphatase activity assay

Phosphatase activity was measured using $p$-nitrophenyl phosphate ($p$-NPP) as the substrate in a final reaction volume of 200 µL. Reactions were initiated by incubating 0–80 µg of purified protein with $p$-NPP at 37°C for 60 min. The release of $p$-nitrophenol ($p$-NP), the dephosphorylated product, was quantified by measuring absorbance at 405 nm using a microplate spectrophotometer.

Kinetic parameters were determined under initial velocity ($V_0$) conditions using substrate concentrations ranging from 0.1 to 10 mM $p$-NPP. The kinetic constants Km and Vmax were calculated by fitting the data to the Michaelis-Menten equation using nonlinear regression analysis in GraphPad Prism 8.0. The catalytic turnover rate (Kcat) was calculated using the equation $Kcat = Vmax/E_0$, where Vmax was expressed as nM $p$-NP min$^{-1}$ and $E_0$ represents the active enzyme concentration. Control reactions were carried out in the absence of either the substrate or the enzyme to measure background absorbance. All assays were performed in triplicate to ensure the reproducibility of results.

## Quantitative real-time PCR

THP-1 cells were infected with the BCG strain and harvested 24 hpi for total RNA isolation. RNA was extracted using 1 mL of TRIzol reagent following the manufacturer's protocol. To eliminate genomic DNA contamination, RNA samples were treated with DNase I for 30 min at 37°C, followed by heat inactivation at 65°C for 10 min. RNA concentration and purity were assessed using a NanoDrop 2000 spectrophotometer (Thermo Fisher Scientific).

For cDNA synthesis, 1 µg of total RNA was reverse transcribed using the HiScript III cDNA Reverse Transcription Kit with random hexamer primers, according to the manufacturer's protocol. qRT-PCR was performed using ChamQ SYBR Green RT-PCR Master Mix. Relative mRNA expression levels were calculated using the $2^{-\Delta\Delta Ct}$ method, normalizing Ct values to the housekeeping gene *HPRT*. All qRT-PCR experiments were conducted in triplicate, with at least three independent biological replicates and two technical replicates per condition. Primer sequences are listed in *Supplementary file 6*.

## RNA-seq analysis

THP-1 cells were seeded at a density of 1×10$^6$ cells well in 6-well plates and differentiated into macrophages by treatment with 200 nM PMA for 48 hr. After differentiation, cells were washed with PBS and infected with BCG WT or ΔMmpE strains at a multiplicity of infection (MOI) of 10:1. Following 24 hr of infection, total RNA was extracted using TRIzol reagent according to the manufacturer's protocol.

High-quality sequencing reads were then aligned to the human reference genome (GRCh38) using HISAT2 (v2.2.1). Transcript quantification was performed using FeatureCounts (v2.0.3), and differential gene expression analysis was conducted using DESeq2 (v1.38.3) in R. Genes with an adjusted $p$-value< 0.05 and an absolute |log$_2$fold change ≥ 1| were considered significantly differentially expressed. RNA-seq data (GEO accession: GSE312039) were visualized using GraphPad Prism version 8.0, as well as volcano plots, heatmaps, and other graphical outputs generated via the SRplot platform (*Tang et al., 2023*).

## CUT&Tag analysis

HEK293T cells were seeded in 10 cm² dishes at a density of 1×10$^7$ cells per well and transfected with 10 µg of the MmpE-pEGFP expression plasmid using Hieff Trans Liposomal Transfection Reagent, following the manufacturer's instructions. Cells transfected with the empty vector pEGFP were used as controls. At 36 hpt, cells were collected for CUT&Tag analysis. For the CUT&Tag (GEO accession: GSE312934), cells were fixed with 1% formaldehyde at room temperature for 10 min, and the reaction was quenched with 0.125 M glycine. After centrifugation, cells were permeabilized using ice-cold wash buffer containing 0.05% digitonin. Primary antibody incubation was carried out overnight at 4°C using mouse anti-GFP antibody (1:1000), followed by a 2 hr incubation at room temperature with

Protein A/G-conjugated secondary antibody (1:100). Cells were then incubated at 37°C for 1 hr with a Protein A-Tn5 transposase complex (1:100) preloaded with Illumina sequencing adapters.

The reaction was terminated using lysis buffer containing 10% SDS, followed by proteinase K digestion. DNA was purified and amplified by PCR (12 cycles) using the NEBNext Ultra II DNA Library Preparation Kit. Fragment size distribution (150–300 bp) was confirmed using the Agilent 2100 Bioanalyzer. Sequencing was performed on the Illumina NovaSeq 6000 platform, producing 150 bp paired-end reads. Raw reads were quality-checked using FastQC and aligned to the human reference genome (hg38). Peak calling was conducted using MACS2 (v2.2.7) with the parameters `--nomodel --qvalue 0.05`. Genomic annotation, GO analysis, and de novo motif discovery were performed using HOMER (v4.11). Identified motifs were matched to known transcription factor binding motifs using TOMTOM, with a statistical significance threshold of $p < 0.001$.

## Chromatin immunoprecipitation (ChIP) assays

ChIP assays were performed to validate the direct binding of MmpE to the promoter regions of target genes. HEK293T cells were transfected with either EGFP or MmpE-EGFP expression plasmids. Cross-linking, chromatin isolation, fragmentation, and immunoprecipitation were carried out following the same procedure described for the CUT&Tag assay. DNA was purified from both input and immuno-precipitated (IP) samples and subjected to PCR using primers specific to the promoter regions of the target genes (–100 bp to +50 bp relative to the transcription start site). *GAPDH* primers were used as a negative control. PCR products were analyzed by agarose gel electrophoresis. For quantitative analysis, ChIP-qPCR was carried out using the same primer sets. All reactions were performed in triplicate. Relative enrichment was calculated using the $2^{-\Delta Ct}$ method with normalization to input DNA.

## Electrophoretic mobility shift assays (EMSA)

DNA substrates were amplified by PCR from the genomic DNA of *Mycobacterium tuberculosis* H37Rv. The sequences of the oligonucleotide fragments used in the assay are listed in *Supplementary file 6*. EMSA was carried out using a modified protocol as previously described (*Li et al., 2020*). Each 20 µL reaction contained PCR-amplified DNA fragments, purified MmpE protein at varying concentrations, and binding buffer composed of 20 mM Tris-HCl (pH 8.0), 50 mM KCl, 50 mM $MgCl_2$, 0.2 mM $NiSO_4$, 0.05 mg/mL bovine serum albumin (BSA), 1 mM DTT, 10% (v/v) glycerol. Reaction mixtures were incubated on ice for 30 min to allow protein-DNA complex formation. Samples were then resolved on a 4.8% native polyacrylamide gel using a running buffer containing 15 mM Tris-HCl (pH 7.5), 0.1 M glycine, 1 mM DTT, 50 µM $MgCl_2$, and 0.2 µM $NiSO_4$ at 80 V. Following electrophoresis, gels were stained with ethidium bromide and visualized using a Gel Imaging Analysis System (JIAPENG, China).

## Detection of phagosomal acidification

THP-1 cells were seeded at 70% confluency in poly-L-lysine-coated 35 mm glass-bottom dishes and maintained in RPMI 1640 supplemented with 10% (v/v) FBS and 200 nM PMA for 24 hr at 37°C under 5% $CO_2$. After two washes with PBS, the cells were infected with DiD-labeled BCG strains at a MOI of 10 for 24 hr, LysoTracker was added for 1 hr, followed by Hoechst staining for 15 min to visualize the nuclei. Hoechst staining was excited at 350 nm and emitted at 461 nm (blue), while LysoTracker was excited at 504 nm and emitted at 511 nm (green); DiD was excited at 644 nm and emitted at 665 nm (red). Image analysis was performed by calculating the percentage co-localization of LysoTracker with the phagosomes to assess the effects of different mutants.

## Colony-forming unit (CFU) assay

For bacterial preparation, mid-log phase cultures of BCG ($OD_{600}=0.6$) were harvested, washed with PBS, and resuspended in fresh medium. THP-1 cells were seeded in 12-well plates at a density of $5 \times 10^5$ cells per well and differentiated into macrophage-like cells by treatment with 200 nM PMA for 24 hr. After differentiation, cells were washed with PBS and treated with the PI3K-Akt-mTOR pathway inhibitor BEZ235 at various concentrations (0–200 nM). BEZ235 was applied with bacterial infection (co-treatment). An equivalent volume of DMSO was used as a vehicle control for all treatment conditions.

Cells were infected at a MOI of 10:1 and incubated for 4 hr. After infection, extracellular bacteria were removed by treatment with penicillin-streptomycin (100 µg/mL) for 4 hr. Cells were then washed

three times with PBS and maintained in antibiotic-free medium until harvest. Intracellular bacterial burdens were assessed at multiple time points post-infection, and macrophages were lysed with 0.25% SDS for 10 min at room temperature. Lysates were serially diluted in PBS, and 100 μL aliquots of each dilution were plated on Middlebrook 7H10 agar supplemented with 10% OADC and kanamycin (50 μg/mL). Plates were incubated at 37°C for 14 days. CFUs were enumerated, and final counts were calculated by applying the corresponding dilution factors. BCG strains used in this study are listed in *Supplementary file 5*.

## Mice infection

Mice were infected as described previously (*Chen et al., 2022*), with minor modifications. Mid-log phase BCG cultures were washed twice in PBS containing 0.05% Tween 80 and briefly sonicated to disrupt bacterial clumps. Female C57BL/6 mice were anesthetized with isoflurane and intranasally inoculated with $1 \times 10^7$ CFUs of BCG in 40 μL of PBS. To confirm the inoculation dose, six mice were euthanized at 48 hpi. Lungs were harvested, homogenized in 1 mL of PBS using a tissue homogenizer, and serial dilutions of the homogenates were plated on Middlebrook 7H10 agar supplemented with 10% OADC. Plates were incubated at 37°C with 5% $CO_2$ for 14 days, after which CFUs were enumerated.

## Histological analysis

Lung tissues from BCG-infected mice were fixed in 4% phosphate-buffered formalin at room temperature for 24 hr, followed by paraffin embedding. Paraffin-embedded tissue samples were sectioned into 2–3 μm thick slices using a microtome. Sections were deparaffinized, rehydrated through a graded ethanol series, and stained with hematoxylin and eosin (H&E). Stained tissue sections were examined using an Olympus BX53 light microscope. For digital analysis and annotation, scanned slides were visualized using CaseViewer software (version 2.0; 3DHISTECH).

## Statistical analyses

Data are presented as mean ± standard deviation (SD). Statistical comparisons were performed using two-tailed unpaired Student's *t*-tests or two-way ANOVA. A *p*-value of less than 0.05 was considered significant, with significance levels indicated as follows: $p<0.05$ (*), $p<0.01$ (**), and $p<0.001$ (***). All statistical analyses were performed using GraphPad Prism (version 8.0). Data were obtained from at least three independent biological replicates, each including two or more technical replicates.

## Acknowledgements

This work was supported by the Fundamental Research Funds for the Central Universities (#2662023DKQD001), Talent Start-up Funds of Huazhong Agricultural University (#11042310008), National Natural Science Foundation of China (#32473123), China Agriculture Research System of MOF and MARA (#CARS-37), and National Key Research and Development Program of China (#2021YFD1800403).

## Additional information

### Funding

| Funder | Grant reference number | Author |
| --- | --- | --- |
| Fundamental Research Funds for Central Universities | 2662023DKQD001 | Lei Zhang |
| Talent Start-up Funds of Huazhong Agricultural University | 11042310008 | Lei Zhang |
| National Natural Science Foundation of China | 32473123 | Aizhen Guo |

| Funder | Grant reference number | Author |
|---|---|---|
| China Agriculture Research System of MOF and MARA | CARS-37 | Aizhen Guo |
| National Key Research and Development Program of China | 2021YFD1800403 | Yingyu Chen |

The funders had no role in study design, data collection and interpretation, or the decision to submit the work for publication.

### Author contributions

Liu Chen, Conceptualization, Data curation, Software, Formal analysis, Validation, Investigation, Visualization, Methodology, Writing – original draft, Writing – review and editing; Baojie Duan, Validation, Investigation; Qiang Jiang, Data curation, Validation, Investigation, Visualization; Yifan Wang, Data curation, Formal analysis, Investigation; Yingyu Chen, Resources, Funding acquisition, Visualization; Lei Zhang, Aizhen Guo, Conceptualization, Resources, Supervision, Funding acquisition, Visualization, Writing – original draft, Project administration, Writing – review and editing

### Author ORCIDs

Liu Chen ⬛ https://orcid.org/0009-0003-3722-6994
Yingyu Chen ⬛ https://orcid.org/0000-0002-1200-5314
Lei Zhang ⬛ https://orcid.org/0000-0002-8566-6068

### Ethics

All animal experiments were conducted in accordance with protocols approved by the Animal Ethics Committee of Huazhong Agricultural University and carried out under the guidelines of the Institutional Animal Care and Use Committee,Permit Number: SYXK (Hubei) 2020-0084.

Reviewer #1 (Public review): https://doi.org/10.7554/eLife.108037.3.sa1
Reviewer #2 (Public review): https://doi.org/10.7554/eLife.108037.3.sa2
Reviewer #3 (Public review): https://doi.org/10.7554/eLife.108037.3.sa3
Author response https://doi.org/10.7554/eLife.108037.3.sa4

# Additional files

### Supplementary files

Supplementary file 1. 175 genes were differentially expressed genes in RNA-seq.

Supplementary file 2. 2903 candidate MmpE-specific ChIP-seq signals.

Supplementary file 3. 298 genes were differentially expressed in both CUTTag and RNA-seq.

Supplementary file 4. Plasmids used in this study.

Supplementary file 5. Bacterial strains used in this study.

Supplementary file 6. Primers used in this study.

MDAR checklist

### Data availability

RNA-seq data were deposited in GEO (accession number GSE312039). CUT&Tag data were deposited in GEO (accession number GSE312934). All data generated or analyzed during this study are included in the manuscript and supporting files; source data files have been provided for all figures.

The following datasets were generated:

| Author(s) | Year | Dataset title | Dataset URL | Database and Identifier |
|---|---|---|---|---|
| Liu C, Duan B, Jiang Q, Wang Y, Chen Y, Hu C, Zhang L, Guo A | 2025 | Differential transcriptional effects of mycobacterial nucleomodulins MgdE and MmpE in THP-1 macrophages | https://www.ncbi.nlm.nih.gov/geo/query/acc.cgi?acc=GSE312039 | NCBI Gene Expression Omnibus, GSE312039 |
| Liu C, Duan B, Jiang Q, Wang Y, Chen Y, Zhang L, Guo A | 2025 | Mycobacterial Metallophosphatase MmpE Functions as a Nucleomodulin to Regulate Host Gene Expression and Promote Intracellular Survival | https://www.ncbi.nlm.nih.gov/geo/query/acc.cgi?acc=GSE312934 | NCBI Gene Expression Omnibus, GSE312934 |

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
