## [Editor Report · eLife Assessment]

The work **convincingly** demonstrates the role of the mycobacterial secreted effector protein MmpE, which translocates to the host nucleus and exhibits phosphatase activity. The study is particularly **valuable** in showing that both the nuclear localization signal sequences and residues critical for phosphatase function are essential for host gene regulation, lysosomal biogenesis, and intracellular survival. Future studies will be needed to explore additional host pathways modulated by MmpE, particularly in the context of infection with a fully virulent *Mycobacterium tuberculosis* strain.

---

## [Referee Report · Reviewer #1 (Public review)]

Summary:

The study provides insightful characterization of the mycobacterial secreted effector protein MmpE which translocates to the host nucleus and exhibits phosphatase activity. The study characterizes the nuclear localization signal sequences and residues critical for the phosphatase activity, both of which are required for intracellular survival

Strengths:

(1) The study addresses the role of nucleomodulins, an understudied aspect in mycobacterial infections.

(2) The authors employ a combination of biochemical and computational analyses along with in vitro and in vivo validations to characterize the role of MmpE.

Weaknesses:

(1) While the study establishes that the phosphatase activity of MmpE operates independently of its NLS, there is a clear gap in understanding how this phosphatase activity supports mycobacterial infection. The investigation lacks experimental data on specific substrates of MmpE or pathways influenced by this virulence factor.

(2) The study does not explore whether the phosphatase activity of MmpE is dependent on the NLS within macrophages, which would provide critical insights into its biological relevance in host cells. Conducting experiments with double knockout/mutant strains and comparing their intracellular survival with single mutants could elucidate these dependencies and further validate the significance of MmpE's dual functions.

(3) The study does not provide direct experimental validation of the MmpE deletion on lysosomal trafficking of the bacteria.

(4) The role of MmpE as a mycobacterial effector would be more relevant using virulent mycobacterial strains such as H37Rv.

Comments on revisions:

I appreciate the work the authors have done to address reviewers comments. The revised manuscript looks significantly improved. My major concern in the revised version is the microscopy data where the BCG staining using the DiD fluorescent stain does not bring out the rod-shaped bacilli structure. I suggest the authors either use a GFP reporter or some other fluorescent stain to address this issue.

---

## [Referee Report · Reviewer #2 (Public review)]

Summary:

In this paper, the authors have characterized Rv2577 as a Fe3+/Zn2+ -dependent metallophosphatase and a nucleomodulin protein. The authors have also identified His348 and Asn359 as critical residues for Fe3+ coordination. The authors show that the proteins encode for two nuclease localization signals. Using C-terminal Flag expression constructs, the authors have shown that MmpE protein is secretory. The authors have prepared genetic deletion strains and show that MmpE is essential for intracellular survival of *M. bovis* BCG in THP-1 macrophages, RAW264.7 macrophages and mice model of infection. The authors have also performed RNA-seq analysis to compare the transcriptional profiles of macrophages infected with wild type and mmpE mutant strain. The relative levels of ~ 175 transcripts were altered in mmpE mutant infected macrophages and majority of these were associated with various immune and inflammatory signalling pathways. Using these deletion strains, the authors proposed that MmpE inhibits inflammatory gene expression by binding to the promoter region of vitamin D receptor. The authors also showed that MmpE arrests phagosome maturation by regulating the expression of several lysosome associated genes such as TFEB, LAMP1, LAMP2 etc. These findings reveal a sophisticated mechanism by which a bacterial effector protein manipulates gene transcription and promotes intracellular survival.

Strength:

The authors have used a combination of cell biology, microbiology and transcriptomics to elucidate the mechanisms by which Rv2577 contributes to intracellular survival.

Weakness:

The authors should thoroughly check the mice data and show individual replicate values in bar graphs.

Comments on revisions:

Thanks to the authors for addressing the concerns raised during the review of the original manuscript. The data is now presented with clarity, and discrepancies in mouse experiments have also been addressed with additional experiments.

---

## [Referee Report · Reviewer #3 (Public review)]

Summary:

In this manuscript titled "Mycobacterial Metallophosphatase MmpE Acts as a Nucleomodulin to Regulate Host Gene Expression and Promote Intracellular Survival", Chen et al describe biochemical characterisation, localisation and potential functions of the gene using a genetic approach in *M. bovis* BCG and perform macrophage and mice infections to understand the roles of this potentially secreted protein in the host cell nucleus. The findings demonstrate the role of a secreted phosphatase of *M. bovis* BCG in shaping the transcriptional profile of infected macrophages, potentially through nuclear localisation and direct binding to transcriptional start sites, thereby regulating the inflammatory response to infection.

Strengths:

The authors demonstrate using a transient transfection method that MmpE when expressed as a GFP-tagged protein in HEK293T cells, exhibits nuclear localisation. The authors identify two NLS motifs that together are required for nuclear localisation of the protein. A deletion of the gene in *M. bovis* BCG results in poorer survival compared to the wild type parent strain, which is also killed by macrophages. Relative to the WT strain infected macrophages, macrophages infected with the mmpE strain exhibited differential gene expression. Overexpression of the gene in HEK293T led to occupancy of the transcription start site of several genes, including the Vitamin D Receptor. Expression of VDR in THP1 macrophages was lower in case of mmpE infection compared to WT infection. This data supports the utility of the overexpression system in identifying potential target loci of MmpE using the HEK293T transfection model. The authors also demonstrate that the protein is a phosphatase and the phosphatase activity of the protein is partially required for bacterial survival but not for regulation of the VDR gene expression.

Weaknesses:

There are significant differences in lysosomal retention between *M. tuberculosis* and *M. bovis* BCG. This study uses BCG and MMPE overexpression to draw conclusions about the impact of the MMPE gene on host gene expression and the bacteria's lysosomal localisation. While the authors have convincingly supported their claims with this model system, the relevance of this mechanism in *M. tuberculosis* infection remains unaddressed.

---

## [Author Response]

The following is the authors’ response to the original reviews.

**Public Reviews:**

**Reviewer #1 (Public review):**
Summary:Review of the manuscript titled " Mycobacterial Metallophosphatase MmpE acts as a nucleomodulin to regulate host gene expression and promotes intracellular survival".The study provides an insightful characterization of the mycobacterial secreted effector protein MmpE, which translocates to the host nucleus and exhibits phosphatase activity. The study characterizes the nuclear localization signal sequences and residues critical for the phosphatase activity, both of which are required for intracellular survival.Strengths:(1) The study addresses the role of nucleomodulins, an understudied aspect in mycobacterial infections.(2) The authors employ a combination of biochemical and computational analyses along with in vitro and in vivo validations to characterize the role of MmpE.Weaknesses:(1) While the study establishes that the phosphatase activity of MmpE operates independently of its NLS, there is a clear gap in understanding how this phosphatase activity supports mycobacterial infection. The investigation lacks experimental data on specific substrates of MmpE or pathways influenced by this virulence factor.

We thank the reviewer for this insightful comment and agree that identification of the substrates of MmpE is important to fully understand its role in mycobacterial infection. MmpE is a putative purple acid phosphatase (PAP) and a member of the metallophosphoesterase (MPE) superfamily. Enzymes in this family are known for their catalytic promiscuity and broad substrate specificity, acting on phosphomonoesters, phosphodiesters, and phosphotriesters (Matange et al., Biochem J, 2015). In bacteria, several characterized MPEs have been shown to hydrolyze substrates such as cyclic nucleotides (e.g., cAMP) (Keppetipola et al., J Biol Chem, 2008; Shenoy et al., J Mol Biol, 2007), nucleotide derivatives (e.g., AMP, UDP-glucose) (Innokentev et al., mBio, 2025), and pyrophosphate-containing compounds (e.g., Ap4A, UDP-DAGn) (Matange et al., Biochem J., 2015). Although the binding motif of MmpE has been identified, determining its physiological substrates remains challenging due to the low abundance and instability of potential metabolites, as well as the limited sensitivity and coverage of current metabolomic technologies in mycobacteria.

(2) The study does not explore whether the phosphatase activity of MmpE is dependent on the NLS within macrophages, which would provide critical insights into its biological relevance in host cells. Conducting experiments with double knockout/mutant strains and comparing their intracellular survival with single mutants could elucidate these dependencies and further validate the significance of MmpE's dual functions.

We thank the reviewer for the comment. Deletion of the NLS motifs did not impair MmpE’s phosphatase activity in vitro (Figure 2F), indicating that MmpE's enzymatic function operates independently of its nuclear localization. Indeed, we confirmed that Fe^3+^-binding ability via the residues H348 and N359 is required for enzymatic activity of MmpE. We have expanded on this point in the Discussion section “MmpE is a bifunctional virulence factor in Mtb”.

(3) The study does not provide direct experimental validation of the MmpE deletion on lysosomal trafficking of the bacteria.

We thank the reviewer for the comment. To validate the role of MmpE in lysosome maturation during infection, we conducted fluorescence colocalization assays in THP-1 macrophages infected with BCG strains, including WT, ∆MmpE, Comp-MmpE, Comp-MmpE^ΔNLS1^, Comp-MmpE^ΔNLS2^, Comp-MmpE^ΔNLS1-2^. These strains were stained with the lipophilic membrane dye DiD, while macrophages were treated with the acidotropic probe LysoTracker Green (Martins et al., Autophagy, 2019). The result indicated that ΔMmpE and MmpE^NLS1-2^ mutants exhibited significantly higher co-localization with LysoTracker compared to WT and Comp-MmpE strains (New Figure 5G), suggesting that MmpE deletion leads to enhanced lysosomal maturation during infection.

(4) The role of MmpE as a mycobacterial effector would be more relevant using virulent mycobacterial strains such as H37Rv.

We thank the reviewer for the comment. Previously, the role of Rv2577/MmpE as a virulence factor has been demonstrated in *M. tuberculosis* CDC 1551, where its deletion significantly reduced bacterial replication in mouse lungs at 30 days post-infection (Forrellad et al., Front Microbiol, 2020). However, that study did not explore the underlying mechanism of MmpE function. In our study, we found that MmpE enhances *M. bovis* BCG survival in macrophages (THP-1 and RAW264.7 both) and in mice (Figure 3, Figure 7A), consistent with its proposed role in virulence. To investigate the molecular mechanism by which MmpE promotes intracellular survival, we used *M. bovis* BCG as a biosafe surrogate and this model is widely accepted for studying mycobacterial pathogenesis (Wang et al., Nat Immunol, 2015; Wang et al., Nat Commun, 2017; Péan et al., Nat Commun, 2017).

**Reviewer #2 (Public review):**
Summary:In this paper, the authors have characterized Rv2577 as a Fe3+/Zn2+ -dependent metallophosphatase and a nucleomodulin protein. The authors have also identified His348 and Asn359 as critical residues for Fe3+ coordination. The authors show that the proteins encode for two nuclease localization signals. Using C-terminal Flag expression constructs, the authors have shown that the MmpE protein is secretory. The authors have prepared genetic deletion strains and show that MmpE is essential for intracellular survival of *M. bovis* BCG in THP-1 macrophages, RAW264.7 macrophages, and a mouse model of infection. The authors have also performed RNA-seq analysis to compare the transcriptional profiles of macrophages infected with wild-type and MmpE mutant strains. The relative levels of ~ 175 transcripts were altered in MmpE mutant-infected macrophages and the majority of these were associated with various immune and inflammatory signalling pathways. Using these deletion strains, the authors proposed that MmpE inhibits inflammatory gene expression by binding to the promoter region of a vitamin D receptor. The authors also showed that MmpE arrests phagosome maturation by regulating the expression of several lysosome-associated genes such as TFEB, LAMP1, LAMP2, etc. These findings reveal a sophisticated mechanism by which a bacterial effector protein manipulates gene transcription and promotes intracellular survival.Strength:The authors have used a combination of cell biology, microbiology, and transcriptomics to elucidate the mechanisms by which Rv2577 contributes to intracellular survival.Weakness:The authors should thoroughly check the mice data and show individual replicate values in bar graphs.

We kindly appreciate the reviewer for the advice. We have now updated the relevant mice data in the revised manuscript.

**Reviewer #3 (Public review):**
Summary:In this manuscript titled "Mycobacterial Metallophosphatase MmpE Acts as a Nucleomodulin to Regulate Host Gene Expression and Promote Intracellular Survival", Chen et al describe biochemical characterisation, localisation and potential functions of the gene using a genetic approach in *M. bovis* BCG and perform macrophage and mice infections to understand the roles of this potentially secreted protein in the host cell nucleus. The findings demonstrate the role of a secreted phosphatase of *M. bovis* BCG in shaping the transcriptional profile of infected macrophages, potentially through nuclear localisation and direct binding to transcriptional start sites, thereby regulating the inflammatory response to infection.Strengths:The authors demonstrate using a transient transfection method that MmpE when expressed as a GFP-tagged protein in HEK293T cells, exhibits nuclear localisation. The authors identify two NLS motifs that together are required for nuclear localisation of the protein. A deletion of the gene in *M. bovis* BCG results in poorer survival compared to the wild-type parent strain, which is also killed by macrophages. Relative to the WT strain-infected macrophages, macrophages infected with the ∆mmpE strain exhibited differential gene expression. Overexpression of the gene in HEK293T led to occupancy of the transcription start site of several genes, including the Vitamin D Receptor. Expression of VDR in THP1 macrophages was lower in the case of ∆mmpE infection compared to WT infection. This data supports the utility of the overexpression system in identifying potential target loci of MmpE using the HEK293T transfection model. The authors also demonstrate that the protein is a phosphatase, and the phosphatase activity of the protein is partially required for bacterial survival but not for the regulation of the VDR gene expression.Weaknesses:(1) While the motifs can most certainly behave as NLSs, the overexpression of a mycobacterial protein in HEK293T cells can also result in artefacts of nuclear localisation. This is not unprecedented. Therefore, to prove that the protein is indeed secreted from BCG, and is able to elicit transcriptional changes during infection, I recommend that the authors (i) establish that the protein is indeed secreted into the host cell nucleus, and (ii) the NLS mutation prevents its localisation to the nucleus without disrupting its secretion.

We kindly appreciate the reviewer for this insightful comment. To confirm the translocation of MmpE into the host nucleus during BCG infection, we first detected the secretion of MmpE by *M. bovis* BCG, using Ag85B as a positive control and GlpX as a negative control (Zhang et al., Nat commun, 2022). Our results showed that MmpE- Flag was present in the culture supernatant, indicating that MmpE is secreted by BCG indeed (new Figure S1C).

Next, we performed immunoblot analysis of the nuclear fractions from infected THP-1 macrophages expressing FLAG-tagged wild-type MmpE and NLS mutants. The results revealed that only wild-type MmpE was detected in the nucleus, while MmpE^ΔNLS1^, MmpE^ΔNLS2^ and MmpE^ΔNLS1-2^ were not detectable in the nucleus (New Figure S1D). Taken together, these findings demonstrated that MmpE is a secreted protein and that its nuclear translocation during infection requires both NLS motifs.

Demonstration that the protein is secreted: Supplementary Figure 3 - Immunoblotting should be performed for a cytosolic protein, also to rule out detection of proteins from lysis of dead cells. Also, for detecting proteins in the secreted fraction, it would be better to use Sauton's media without detergent, and grow the cultures without agitation or with gentle agitation. The method used by the authors is not a recommended protocol for obtaining the secreted fraction of mycobacteria.

We kindly appreciate the reviewer for the advice. To avoid the effects of bacterial lysis, we cultured the BCG strains expressing MmpE-Flag in Middlebrook 7H9 broth with 0.5% glycerol, 0.02% Tyloxapol, and 50 µg/mL kanamycin at 37 °C with gentle agitation (80 rpm) until an OD_600_ of approximately 0.6 (Zhang et al., Nat Commun, 2022). Subsequently, we assessed the secretion of MmpE-Flag in the culture supernatant, using Ag85B as a positive control and GlpX as a negative control (New Figure S1C). The results showed that GlpX was not detected in the supernatant, while MmpE and Ag85B were detected, indicating that MmpE is indeed a secreted protein in BCG.

Demonstration that the protein localises to the host cell nucleus upon infection: Perform an infection followed by immunofluorescence to demonstrate that the endogenous protein of BCG can translocate to the host cell nucleus. This should be done for an NLS1-2 mutant expressing cell also.

We thank the reviewer for the suggestion. We agree that this experiment would be helpful to further verify the ability of MmpE for nuclear import. However, MmpE specific antibody is not available for us for immunofluorescence experiment. Alternatively, we performed nuclear-cytoplasmic fractionation for the THP-1 cells infected with the *M. bovis* BCG strains expressing FLAG-tagged wild-type MmpE, as well as NLS deletion mutants (MmpE^ΔNLS1^, MmpE^ΔNLS2^, and MmpE^ΔNLS1-2^). The WT MmpE is detectable in both cytoplasmic and nuclear compartments, while MmpE^ΔNLS1^, MmpE^ΔNLS2^ or MmpE^ΔNLS1-2^ were almost undetectable in nuclear fractions (New Figure S1D), suggesting that both NLS motifs are necessary for nuclear import.

(2) In the RNA-seq analysis, the directionality of change of each of the reported pathways is not apparent in the way the data have been presented. For example, are genes in the cytokine-cytokine receptor interaction or TNF signalling pathway expressed more, or less in the ∆mmpE strain?

We thank the reviewer for the comment. The KEGG pathway enrichment diagrams in our RNA-seq analysis primarily reflect the statistical significance of pathway enrichment based on differentially expressed genes, but do not indicate the directionality of genes expression changes. To address this concern, we conducted qRT-PCR on genes associated with the cytokine-cytokine receptor interaction pathway, specifically *IL23A*, *CSF2*, and *IL12B*. The results showed that, compared to the WT strain, infection with the ΔMmpE strain resulted in significantly increased expression levels of these genes in THP-1 cells (Figure 4F, Figure S4B), consistent with the RNA-seq data. Furthermore, we have submitted the complete RNA-seq dataset to the NCBI GEO repository [GSE312039], which includes normalized expression values and differential expression results for all detected genes.

(3) Several of these pathways are affected as a result of infection, while others are not induced by BCG infection. For example, BCG infection does not, on its own, produce changes in IL1β levels. As the author s did not compare the uninfected macrophages as a control, it is difficult to interpret whether ∆mmpE induced higher expression than the WT strain, or simply did not induce a gene while the WT strain suppressed expression of a gene. This is particularly important because the strain is attenuated. Does the attenuation have anything to do with the ability of the protein to induce lysosomal pathway genes? Does induction of this pathway lead to attenuation of the strain? Similarly, for pathways that seem to be downregulated in the ∆mmpE strain compared to the WT strain, these might have been induced upon infection with the WT strain but not sufficiently by the ∆mmpE strain due to its attenuation/ lower bacterial burden.

We thank the reviewer for the comment. Previous studies have shown that wild-type BCG induces relatively low levels of IL-1β, while retaining partial capacity to activate the inflammasome (Qu et al., Sci Adv, 2020). Our data (Figures 3G) show that infection with the ΔMmpE strain results in enhanced IL-1β expression, consistent with findings by Master et al. (Cell Host Microbe, 2008), in which deletion of *zmp1* in BCG or *M. tuberculosis* led to increased IL-1β levels due to reduced inhibition of inflammasome activation.

In the revised manuscript, we have provided additional qRT-PCR data using uninfected macrophages as a baseline control. These results demonstrate that the WT strain suppresses lysosome-associated gene expression, whereas the ΔMmpE strain upregulates these genes, indicating that MmpE inhibits lysosome-related genes expression (Figure 4G). Furthermore, bacterial burden analysis revealed that ∆*mmpE* exhibited ~3-fold lower intracellular survival than the WT strain in THP-1 cells. However, when lysosomal maturation was inhibited, the difference in bacterial load between the two strains was reduced to ~1-fold (New Figures S6B and C). These findings indicate that MmpE promotes intracellular survival primarily by inhibiting lysosomal maturation, which is consistent with a previous study (Chandra et al., Sci Rep, 2015).

(4) CHIP-seq should be performed in THP1 macrophages, and not in HEK293T. Overexpression of a nuclear-localised protein in a non-relevant line is likely to lead to several transcriptional changes that do not inform us of the role of the gene as a transcriptional regulator during infection.

We thank the reviewer for the comment. We performed ChIP-seq in HEK293T cells based on their high transfection efficiency, robust nuclear protein expression, and well-annotated genome (Lampe et al., Nat Biotechnol, 2024; Marasco et al., Cell, 2022). These characteristics make HEK293T an ideal system for the initial identification of genome-wide chromatin binding profiles by MmpE.

Further, we performed comprehensive validation of the ChIP-seq findings in THP-1 macrophages. First, CUT&Tag and RNA-seq analyses in THP-1 cells revealed that MmpE modulates genes involved in the PI3K–AKT signaling and lysosomal maturation pathways (Figure 4C; Figure S5A-B). Correspondingly, we found that infection with the ΔMmpE strain led to reduced phosphorylation of AKT (S473), mTOR (S2448), and p70S6K (T389) (New Figure 5E-F), and upregulation of lysosomal genes such as TFEB, LAMP1, and LAMP2 (Figure 4G), compared to infection with the WT strain, and lysosomal maturation in cells infected with the ΔMmpE strain more obviously (New Figure 5G). Additionally, CUT&Tag profiling identified MmpE binding at the promoter region of the *VDR* gene, which was further validated by EMSA and ChIP-qPCR. Also, qRT-PCR demonstrated that MmpE suppresses VDR transcription, supporting its role as a transcriptional regulator (Figure 6). Collectively, these data confirm the biological relevance and functional significance of the ChIP-seq findings obtained in HEK293T cells.

(5) I would not expect to see such large inflammatory reactions persisting 56 days post-infection with *M. bovis* BCG. Is this something peculiar for an intratracheal infection with 1x107 bacilli? For images of animal tissue, the authors should provide images of the entire lung lobe with the zoomed-in image indicated as an inset.

We thank the reviewer for the comment. The lung inflammation peaked at days 21–28 and had clearly subsided by day 56 across all groups (New Figure 7B), consistent with the expected resolution of immune responses to an attenuated strain like *M. bovis* BCG. This temporal pattern is in line with previous studies using intravenous or intratracheal BCG vaccination in mice and macaques, which also demonstrated robust early immune activation followed by resolution over time (Smith et al., Nat Microbiol, 2025; Darrah et al., Nature, 2020).

In this study, the infectious dose (1×10^7^ CFU intratracheal) was selected based on previous studies in which intratracheal delivery of 1×10^7^ CFU produced consistent and measurable lung immune responses and pathology without causing overt illness or mortality (Xu et al., Sci Rep, 2017; Niroula et al., Sci Rep, 2025). We have provided whole-lung lobe images with zoomed-in insets in the source dataset.

(6) For the qRT-PCR based validation, infections should be performed with the MmpE-complemented strain in the same experiments as those for the WT and ∆mmpE strain so that they can be on the same graph, in the main manuscript file. Supplementary Figure 4 has three complementary strains. Again, the absence of the uninfected, WT, and ∆mmpE infected condition makes interpretation of these data very difficult.

We thank the reviewer for the comment. As suggested, we have conducted the qRT-PCR experiment including the uninfected, WT, ∆mmpE, Comp-MmpE, and the three complementary strains infecting THP-1 cells (Figure 4F and G; New Figure S4B–D).

(7) The abstract mentions that MmpE represses the PI3K-Akt-mTOR pathway, which arrests phagosome maturation. There is not enough data in this manuscript in support of this claim. Supplementary Figure 5 does provide qRT-PCR validation of genes of this pathway, but the data do not indicate that higher expression of these pathways, whether by VDR repression or otherwise, is driving the growth restriction of the ∆mmpE strain.

We thank the reviewer for the comment. In the updated manuscript, we have provided more evidence. First, the RNA-seq analysis indicated that MmpE affects the PI3K-AKT signaling pathway (Figure 4C). Second, CUT&Tag analysis suggested that MmpE binds to the promoter regions of key pathway components, including *PRKCB*, *PLCG2*, and *PIK3CB* (Figure S5A). Third, confocal microscopy showed that ΔMmpE strain promotes significantly increased lysosomal maturation compared to the WT, a process downstream of the PI3K-AKT-mTOR axis (New Figure 5G).

Further, we measured protein phosphorylation for validating activation of the pathway (Zhang et al., Stem Cell Reports, 2017). Our results showed that cells infected with WT strains exhibited significantly higher phosphorylation of Akt, mTOR, and p70S6K compared to those infected with ΔMmpE strains (New Figures 5E and F). Moreover, the dual PI3K/mTOR inhibitor BEZ235 abolished the survival advantage of WT strains over ΔMmpE mutants in THP-1 macrophages (New Figure S6B and C). Collectively, these results support that MmpE activates the PI3K–Akt–mTOR signaling pathway to enhance bacterial survival within the host.

(8) The relevance of the NLS and the phosphatase activity is not completely clear in the CFU assays and in the gene expression data. Firstly, there needs to be immunoblot data provided for the expression and secretion of the NLS-deficient and phosphatase mutants. Secondly, CFU data in Figure 3A, C, and E must consistently include both the WT and ∆mmpE strain.

We thank the reviewer for the comment. We have now added immunoblot analysis for expression and secretion of MmpE mutants. The result show that NLS-deficient and phosphatase mutants can detected in supernatant (New Figure S1C). Additionally, we have revised Figures 3A, 3C, and 3E to consistently include both the WT and ΔMmpE strains in the CFU assays (Figures 3A, 3C, and 3E).

**Recommendations for the authors:**

**Reviewer #2 (Recommendations for the authors):**
The authors should attempt to address the following comments:(1) Please perform densitometric analysis for the western blot shown in Figure 1E.

We sincerely thank the reviewer for the suggestion. In the updated manuscript, we have performed densitometric analysis of the western blot shown in New Figure 1F and G.

(2) Is it possible to measure the protein levels for MmpE in lysates prepared from infected macrophages.

We thank the reviewer for the comment. In the revised manuscript, we performed immunoblot analysis to measure MmpE levels in lysates from infected macrophages. The results demonstrated that wild-type MmpE was present in both the cytoplasmic and nuclear fractions during infection in THP-1 cells (New Figure S1D).

(3) The authors should perform circular dichroism studies to compare the secondary structure of wild type and mutant proteins (in particular MmpEHis348 and MmpEAsn359).

We thank the reviewer for this valuable suggestion. We agree that circular dichroism spectroscopy could provide useful information in comparison of the differences on the secondary structures. However, due to the technical limitations, we instead compared the structures of wild-type MmpE and the His348 and Asn359 mutant proteins predicted by AlphaFold. These structural models showed almost no differences in secondary structures between the wild-type and mutants (Figure S1B).

(4) The authors should perform more experiments to determine the binding motif for MmpE in the promoter region of VDR.

We thank the reviewer for this suggestion. In the current study, we have identified the MmpE-binding motif within the promoter region of VDR using CUT&Tag sequencing. This prediction was further validated by ChIP-qPCR and EMSA (Figure 6). These complementary approaches collectively support the identification of a specific MmpE-binding motif and demonstrate its functional relevance. Such approach was acceptable in many publications (Wen et al., Commun Biol, 2020; Li et al., Nat Commun, 2022).

(5) Were the transcript levels of VDR also measured in the lung tissues of infected animals?

We thank the reviewer for this suggestion. In the revised manuscript, we have performed qRT-PCR to assess VDR transcript levels in the lung tissues of infected mice (New Figure S8B).

(6) How does MmpE regulate the expression of lysosome-associated genes?

We thank the reviewer for this question. Our experiments suggested that MmpE suppresses lysosomal maturation probably by activating the host PI3K–AKT–mTOR signaling pathway (New Figure 5E–I). This pathway is well established as a negative regulator of lysosome biogenesis and function (Yang et al., Signal Transduct Target Ther, 2020; Cui et al., Nature, 2023; Cui et al., Nature, 2025). During infection, THP-1 cells infected with the WT showed increased phosphorylation of Akt, mTOR, and p70S6K compared to those infected with ΔMmpE (New Figure S5C, New Figure 5E and F), and concurrently downregulated key lysosomal maturation markers, including TFEB, LAMP1, LAMP2, and multiple V-ATPase subunits (Figure 4G). Given that PI3K–AKT–mTOR signaling suppresses TFEB activity and lysosomal gene transcription (Palmieri et al., Nat Commun, 2017), we propose that MmpE modulates lysosome-associated gene expression and lysosomal function probably by PI3K–AKT–mTOR signaling pathway.

(7) Mice experiment:(a) The methods section states that mice were infected intranasally, but the legend for Figure 6 states intratracheally. Kindly check?(b) Supplementary Figure 7 - this is not clear. The legend says bacterial loads in spleens (CFU/g) instead of DNA expression, as shown in the figure.(c) The data in Figure 6 and Figure S7 seem to be derived from the same experiment, but the number of animals is different. In Figure 6, it is n = 6, and in Figure S7, it is n=3.

We thank the reviewer for the comments.

(a) The infection was performed intranasally, and the figure legend for New Figure 7 has now been corrected.

(b) We adopted quantitative PCR method to measure bacterial DNA levels in the spleens of infected mice. We have now revised the legend.

(c) We have conducted new experiments where each experiment now includes six mice. The results are showed in Figure 7B and C, as well as in the new Figure S8.

(8) The authors should show individual values for various replicates in bar graphs (for all figures).

We thank the reviewer for this helpful suggestion. We have now updated all relevant bar graphs to include individual data points for each biological replicate.

(9) The authors should validate the relative levels of a few DEGs shown in Figure 3F, Figure 3G, and Figure S4C, in the lung tissues of mice infected with wild-type, mutant, and complemented strains.

We thank the reviewer for this suggestion. In the revised manuscript, we have performed qRT-PCR to validate the expression levels of selected DEGs, including inflammation-related and lysosome-associated genes, in lung tissues from mice infected with wild-type, mutant, and complemented strains (New Figure S8C-H).

(10) Did the authors perform an animal experiment using a mutant strain complemented with the phosphatase-deficient MmpE (Comp-MmpE-H348AN359H)?

We appreciate the reviewer's comment. We agree that an additional animal experiment would be useful to assess the effects of the phosphatase. However, our study mainly focused on interpreting the function of the nuclear localization of MmpE during BCG infection. Additionally, we have assessed the role of the phosphatase of MmpE during infection with cell model (Figure 3E).

Minor comment:The mutant strain should be verified by either Southern blot or whole genome sequencing.

We thank the reviewer for this comment. We verified deletion of *mmpE* gene by PCR method (Figure S3A-D) which was acceptable in many publications (Zhang et al., PLoS Pathog, 2020; Zhang et al., Nat Commun, 2022).

**Reviewer #3 (Recommendations for the authors):**
(1) Line 195: cytokine.

We thank the reviewer for the comments. We have now corrected it.

(2) Line 225: rewording required.

Corrected.

(3) Figure 4A. "No difference" instead of "No different".

Corrected.

(4) "KommpE" should be replaced with "∆mmpE strain" (∆=delta symbol).

Corrected.

(5) Supplementary Figure 7. The figure legend states CFU assays, but the y-axis and the graph seem to depict IS1081 quantification.

We thank the reviewer for the comment. The figure is based on IS1081 quantification using qRT-PCR, not CFU assays. We have now revised the legend for New Figure S8A.

References

Chandra P, Ghanwat S, Matta SK, Yadav SS, Mehta M, Siddiqui Z, Singh A, Kumar D (2015) *Mycobacterium tuberculosis* Inhibits RAB7 Recruitment to Selectively Modulate Autophagy Flux in Macrophages Sci Rep 5:16320.

Darrah PA, Zeppa JJ, Maiello P, Hackney JA, Wadsworth MH 2nd, Hughes TK, Pokkali S, Swanson PA 2nd, Grant NL, Rodgers MA, Kamath M, Causgrove CM, Laddy DJ, Bonavia A, Casimiro D, Lin PL, Klein E, White AG, Scanga CA, Shalek AK, Roederer M, Flynn JL, Seder RA (2020) Prevention of tuberculosis in macaques after intravenous BCG immunization Nature 577:95-102.

Forrellad MA, Blanco FC, Marrero Diaz de Villegas R, Vázquez CL, Yaneff A, García EA, Gutierrez MG, Durán R, Villarino A, Bigi F (2020) Rv2577 of *Mycobacterium tuberculosis* Is a virulence factor with dual phosphatase and phosphodiesterase functions Front Microbiol 11:570794.

Innokentev A, Sanchez AM, Monetti M, Schwer B, Shuman S (2025) Efn1 and Efn2 are extracellular 5'-nucleotidases induced during the fission yeast response to phosphate starvation mBio 16: e0299224.

Keppetipola N, Shuman S (2008) A phosphate-binding histidine of binuclear metallophosphodiesterase enzymes is a determinant of 2',3'-cyclic nucleotide phosphodiesterase activity J Biol Chem 283:30942-9.

Lampe GD, King RT, Halpin-Healy TS, Klompe SE, Hogan MI, Vo PLH, Tang S, Chavez A, Sternberg SH (2024) Targeted DNA integration in human cells without double-strand breaks using CRISPR-associated transposases Nat Biotechnol 42:87-98.

Li Z, Sheerin DJ, von Roepenack-Lahaye E, Stahl M, Hiltbrunner A (2022) The phytochrome interacting proteins ERF55 and ERF58 repress light-induced seed germination in *Arabidopsis thaliana* Nat Commun 13:1656.

Marasco LE, Dujardin G, Sousa-Luís R, Liu YH, Stigliano JN, Nomakuchi T, Proudfoot NJ, Krainer AR, Kornblihtt AR (2022) Counteracting chromatin effects of a splicing-correcting antisense oligonucleotide improves its therapeutic efficacy in spinal muscular atrophy Cell 185:2057-2070.e15.

Martins WK, Santos NF, Rocha CS, Bacellar IOL, Tsubone TM, Viotto AC, Matsukuma AY, Abrantes ABP, Siani P, Dias LG, Baptista MS (2019) Parallel damage in mitochondria and lysosomes is an efficient way to photoinduce cell death Autophagy 15:259-279.

Master SS, Rampini SK, Davis AS, Keller C, Ehlers S, Springer B, Timmins GS, Sander P, Deretic V (2008) *Mycobacterium tuberculosis* prevents inflammasome activation Cell Host Microbe 3:224-32.

Matange N, Podobnik M, Visweswariah SS (2015) Metallophosphoesterases: structural fidelity with functional promiscuity Biochem J 467:201-16.

Niroula N, Ghodasara P, Marreros N, Fuller B, Sanderson H, Zriba S, Walker S, Shury TK, Chen JM (2025) Orally administered live BCG and heat-inactivated *Mycobacterium bovis* protect bison against experimental bovine tuberculosis Sci Rep 15:3764.

Palmieri M, Pal R, Nelvagal HR, Lotfi P, Stinnett GR, Seymour ML, Chaudhury A, Bajaj L, Bondar VV, Bremner L, Saleem U, Tse DY, Sanagasetti D, Wu SM, Neilson JR, Pereira FA, Pautler RG, Rodney GG, Cooper JD, Sardiello M (2017) mTORC1-independent TFEB activation via Akt inhibition promotes cellular clearance in neurodegenerative storage diseases Nat Commun 8:14338.

Péan CB, Schiebler M, Tan SW, Sharrock JA, Kierdorf K, Brown KP, Maserumule MC, Menezes S, Pilátová M, Bronda K, Guermonprez P, Stramer BM, Andres Floto R, Dionne MS (2017) Regulation of phagocyte triglyceride by a STAT-ATG2 pathway controls mycobacterial infection Nat Commun 8:14642.

Qu Z, Zhou J, Zhou Y, Xie Y, Jiang Y, Wu J, Luo Z, Liu G, Yin L, Zhang XL (2020) Mycobacterial EST12 activates a RACK1-NLRP3-gasdermin D pyroptosis-IL-1β immune pathway Sci Adv 6: eaba4733.

Shenoy AR, Capuder M, Draskovic P, Lamba D, Visweswariah SS, Podobnik M (2007) Structural and biochemical analysis of the Rv0805 cyclic nucleotide phosphodiesterase from *Mycobacterium tuberculosis* J Mol Biol 365:211-25.

Smith AA, Su H, Wallach J, Liu Y, Maiello P, Borish HJ, Winchell C, Simonson AW, Lin PL, Rodgers M, Fillmore D, Sakal J, Lin K, Vinette V, Schnappinger D, Ehrt S, Flynn JL (2025) A BCG kill switch strain protects against *Mycobacterium tuberculosis* in mice and non-human primates with improved safety and immunogenicity Nat Microbiol 10:468-481.

Wang J, Ge P, Qiang L, Tian F, Zhao D, Chai Q, Zhu M, Zhou R, Meng G, Iwakura Y, Gao GF, Liu CH (2017) The mycobacterial phosphatase PtpA regulates the expression of host genes and promotes cell proliferation Nat Commun 8:244.

Wang J, Li BX, Ge PP, Li J, Wang Q, Gao GF, Qiu XB, Liu CH (2015) *Mycobacterium tuberculosis* suppresses innate immunity by coopting the host ubiquitin system Nat Immunol 16:237–245

Wen X, Wang J, Zhang D, Ding Y, Ji X, Tan Z, Wang Y (2020) Reverse Chromatin Immunoprecipitation (R-ChIP) enables investigation of the upstream regulators of plant genes Commun Biol 3:770.

Xu X, Lu X, Dong X, Luo Y, Wang Q, Liu X, Fu J, Zhang Y, Zhu B, Ma X (2017) Effects of hMASP-2 on the formation of BCG infection-induced granuloma in the lungs of BALB/c mice Sci Rep 7:2300.

Zhang L, Hendrickson RC, Meikle V, Lefkowitz EJ, Ioerger TR, Niederweis M. (2020) Comprehensive analysis of iron utilization by *Mycobacterium tuberculosis* PLoS Pathog 16: e1008337.

Zhang L, Kent JE, Whitaker M, Young DC, Herrmann D, Aleshin AE, Ko YH, Cingolani G, Saad JS, Moody DB, Marassi FM, Ehrt S, Niederweis M (2022) A periplasmic cinched protein is required for siderophore secretion and virulence of *Mycobacterium tuberculosis* Nat Commun 13:2255.

Zhang X, He X, Li Q, Kong X, Ou Z, Zhang L, Gong Z, Long D, Li J, Zhang M, Ji W, Zhang W, Xu L, Xuan A (2017) PI3K/AKT/mTOR Signaling Mediates Valproic Acid-Induced Neuronal Differentiation of Neural Stem Cells through Epigenetic Modifications Stem Cell Reports 8:1256-1269.